# Shapify: Paths to SARS-CoV-2 frameshifting pseudoknot

**Luke Trinity**[1]*, **Ian Wark**[2], **Lance Lansing**[1], **Hosna Jabbari**[1,2,3‡]*, **Ulrike Stege**[1‡]*

**1** Department of Computer Science, University of Victoria, Victoria, British Columbia, Canada, **2** Department of Computing Science, University of Alberta, Edmonton, Alberta, Canada, **3** Institute on Aging and Lifelong Health, Victoria, British Columbia, Canada

‡These authors are joint senior authors on this work.
* ltrinity@uvic.ca (LT); jabbari@uvic.ca (HJ); ustege@uvic.ca (US)

**Data Availability Statement:** The datasets generated and/or analysed during the current study are available in the frameshifting repository: github. com/HosnaJabbari/frameshifting. Shapify is

## Abstract

Multiple coronaviruses including MERS-CoV causing Middle East Respiratory Syndrome, SARS-CoV causing SARS, and SARS-CoV-2 causing COVID-19, use a mechanism known as −1 programmed ribosomal frameshifting (−1 PRF) to replicate. SARS-CoV-2 possesses a unique RNA pseudoknotted structure that stimulates −1 PRF. Targeting −1 PRF in SARS-CoV-2 to impair viral replication can improve patients' prognoses. Crucial to developing these therapies is understanding the structure of the SARS-CoV-2 −1 PRF pseudoknot. Our goal is to expand knowledge of −1 PRF structural conformations. Following a structural alignment approach, we identify similarities in −1 PRF pseudoknots of SARS-CoV-2, SARS-CoV, and MERS-CoV. We provide in-depth analysis of the SARS-CoV-2 and MERS-CoV −1 PRF pseudoknots, including reference and noteworthy mutated sequences. To better understand the impact of mutations, we provide insight on −1 PRF pseudoknot sequence mutations and their effect on resulting structures. We introduce *Shapify*, a novel algorithm that given an RNA sequence incorporates structural reactivity (SHAPE) data and partial structure information to output an RNA secondary structure prediction within a biologically sound hierarchical folding approach. Shapify enhances our understanding of SARS-CoV-2 −1 PRF pseudoknot conformations by providing energetically favourable predictions that are relevant to structure-function and may correlate with −1 PRF efficiency. Applied to the SARS-CoV-2 −1 PRF pseudoknot, Shapify unveils previously unknown paths from initial stems to pseudoknotted structures. By contextualizing our work with available experimental data, our structure predictions motivate future RNA structure-function research and can aid 3-D modeling of pseudoknots.

## Author summary

Identifying inter-viral structural similarity in frameshifting pseudoknots is valuable for treatment development as existing viruses mutate or novel diseases emerge. Computational methods for RNA secondary structure prediction remain critical tools for understanding functional structures, and advancing knowledge for the development of possible treatment options. However, current understanding of the full landscape of potential

available at github.com/ltrinity/Shapify (DOI: 10.5281/zenodo.6100185).

**Funding:** Funding for LT is provided by a University of Victoria Graduate Fellowship, and the Natural Sciences and Engineering Research Council Discovery Grants to HJ, US. Finally, Microsoft AI4Health Azure Grant to HJ enabled data collection via cloud computing. The funders had no role in study design, data collection and analysis, decision to publish, or preparation of the manuscript.

**Competing interests:** The authors have declared that no competing interests exist.

structures for any given RNA is limited by the cost and complexity of experimental methods and intractability of comprehensive algorithmic approaches.

We followed a structural alignment approach to identify the consensus structure for SARS-CoV and MERS-CoV −1 PRF pseudoknots, with SARS-CoV-2 as reference. We developed Shapify for improved prediction of possibly pseudoknotted RNA secondary structures guided by SHAPE reactivity data. Then, we shed light on the structure formation of the SARS-CoV-2 and MERS-CoV frameshifting pseudoknots by analyzing possible structural conformations. Previous structural predictions obtained via different methods all have significant differences. Our results demonstrate innate resiliency via converging paths of the SARS-CoV-2 virus in achieving the native −1 PRF stimulating pseudoknot. Fully accounting for pan-coronaviral structural conformations, which include transient and suboptimal structures, is vital for comprehending viral function.

## Introduction

Sequence analyses of the SARS-CoV-2 genome classify it as a member of the Betacoronavirus subfamily, which includes SARS-CoV and MERS-CoV [1, 2]. All coronaviruses use a particular replication strategy called ribosomal frameshifting, which is a promising target for therapeutic drug development [3]. They utilize the combination of a *slippery sequence* (where the ribosome is prone to slipping forwards or backwards into a different reading frame) and an *RNA pseudoknot* to cause the ribosome to shift from one reading frame into the other [4, 5]. The expected structures of −1 PRF stimulating pseudoknots, referred to as *native* or *wild-type* pseudoknots, have been identified and studied for multiple viruses [6, 7]. In addition, it was found that some non-native pseudoknot conformations, those that differ from the native structure, play a role in regulating frameshifting [6]. Specifically, the *conformational plasticity* of pseudoknots, i.e., their ability to form non-native structures, was established to be correlated with −1 PRF efficiency [6].

In the case of SARS-CoV-2, two different long open reading frames comprise two-thirds of the viral genome: ORF1a and ORF1b (cf. Fig 1) [8]. ORF1b is out of frame with respect to ORF1a, meaning the ribosome will not translate both frames without shifting from one frame to the other.

The native structure of the frameshift stimulating pseudoknot in SARS-CoV-2 possesses a unique three-stemmed structure, an H-type pseudoknot (cf. Fig 2). Kelly et al. [9] observed the rate of frameshifting in SARS-CoV-2 to be approximately 15%-30%, indicating there is further variation in structure beyond what the authors could capture in their experimental procedure.

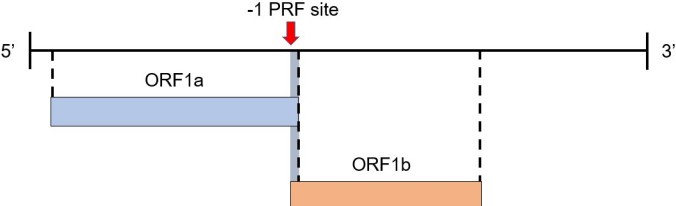

**Fig 1. SARS-CoV-2 viral genome.** −1 PRF site marked by the red arrow. Normally, ribosome translates the complete ORF1a. Sometimes at the −1 PRF site, ribosome shifts from ORF1a to ORF1b, resulting in synthesis of fusion ORF1a/1b polypeptide.

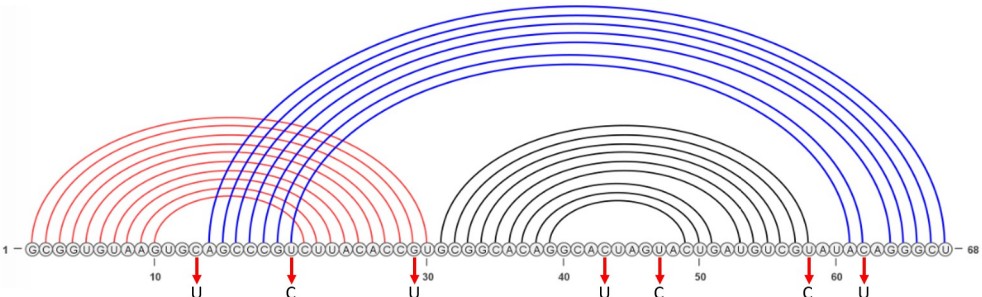

**Fig 2. SARS-CoV-2 −1 PRF pseudoknot native structure.** Stem 1 in red, Stem 2 in black, and pseudoknotted Stem 3 in blue. Visualization generated using VARNA [11]. Mutations of interest shown with red arrows.

Indeed, unfolding experiments to investigate structural dynamics of the SARS-CoV-2 frameshifting pseudoknot revealed multiple distinct conformations [10].

The structural model proposed for the SARS-CoV-2 −1 PRF pseudoknot continues to evolve. Huston et al. [12] found complex folding dynamics with multiple conformational clusters. In [13–15], additional structures for the pseudoknot have been proposed. Furthermore, Yan et al. [16] suggest generalized structure motifs that may be affected by unfolding dynamics during translocation. However, there has not yet been a focus on identifying the ensemble of energetically favourable structures in proximity to the minimum free energy structure of the pseudoknot. Given that non-native folding paths correlate with −1 PRF efficiency and are relevant to structure function, further research on identifying energetically favourable SARS-CoV-2 −1 PRF pseudoknot conformations is valuable in structural prediction.

Most of the existing software packages to determine 3-D conformations of RNA utilize *secondary structure* (set of base pairs) as an input constraint [17–22]. Indeed, Omar et al. [23] used the known native secondary structure as a starting point in their 3-D physics-based modeling of the SARS-CoV-2 −1 PRF pseudoknot and subsequently identified three unique stable conformations. The non-native secondary structures of the SARS-CoV-2 −1 PRF pseudoknot, however, have yet to be identified and incorporated into structural prediction efforts. Here, we provide alternate starting points and input constraints, which can improve both the accuracy and interpretability of 3-D physics-based modeling of the pseudoknot.

Mutations have been reported in all components of the SARS-CoV-2 −1 PRF pseudoknot sequences [24, 25] available on GenBank [26] and GISAID [27]. Although there are far less available sequenced genomes for MERS-CoV, mutations similar to the ones found in SARS-CoV-2 were observed in the MERS-CoV −1 PRF pseudoknot sequence in multiple samples [28]. Yet, the possible effects of such mutations on RNA structure formation and resulting secondary structures have not been fully characterized. Understanding the impact of mutations is vital to long-term success of treatments that rely on RNA structure. Interestingly, the most prevalent mutation to the SARS-CoV-2 −1 PRF pseudoknot sequence increased its similarity to the MERS-CoV −1 PRF pseudoknot [24]. Further validation of this similarity between the SARS-CoV-2 and MERS-CoV −1 PRF pseudoknots is highly relevant for future coronavirus structure-function prediction efforts.

Here, we computationally explore how the single mutation observed in the MERS-CoV −1 PRF pseudoknot sequence changes structure prediction. Kelly et al. [9] evaluated effects of 14 mutations aiming to experimentally disrupt −1 PRF function of SARS-CoV-2. We seek to expand their work by assessing single nucleotide (nt) mutations observed in the general population that were not considered in their work. A single nucleotide change from cytosine to uracil at position 62 (C62U) was identified in a significant proportion of sequences [24]. Based on

information in the COVID-19 CG database [29], which contains over 10.2 million sequences via GISAID [27] as of September 2022, C62U continues to be the most prevalent mutation in the −1 PRF pseudoknot sequence (0.33% of global sequences). Only two other mutations in the −1 PRF pseudoknot sequence were detected in over 0.1% of global samples: A8G in 0.25%, and C43U in 0.16% of all sequences. Notably, no mutations have been detected in the −1 PRF pseudoknot sequences of any of the following lineages: B.1.1.7 (a.k.a. 20I/501Y.V1 Variant of Concern), B.1.351 (a.k.a. 501.V2 variant, 20C/501Y.V2), B.1.617.2 (a.k.a. Variant of Concern Delta G/478K.V1), and B.1.1.52A+BA.* lineage (a.k.a. Variant of Concern Omicron GRA) including sublineages (i.e., B.A.2, B.A.3, B.A.4 and B.A.5 [30, 31]) see Table S in S1 File for accession IDs.

Even a single nucleotide change in an RNA sequence can affect the resulting structure, with functional implications. There is evidence that as little as one or two mutations or deletions can completely disrupt the native structure formation of the SARS-CoV-2 frameshifting pseudoknot [32, 33]. Neupane et al. [25] identified that a uracil to cytosine mutation (U20C) for the SARS-CoV-2 −1 PRF pseudoknot resulted in more than a three-fold reduction in −1 PRF efficiency [25]. We further explore the mutations presented in [25], as these mutations occur in regions important for the formation of the −1 PRF pseudoknot. Three of these mutations are located near the junction of stems (U20C, G29U, U58C), one is adjacent to an adenine bulge identified as critical in frameshifting (C62U) [34], and two are in Loop 2 where mutations have been shown to reduce frameshifting efficiency in SARS-CoV (C43U, U47C) [35]. Incorporating known mutations in pseudoknot structure prediction will advance the understanding of potential conformations, which could alter frameshifting efficiency with implications for viral infectivity and pathology.

It was noted that reducing frameshifting efficiency of SARS-CoV attenuated viral propagation to a significant degree [34, 36, 37]. Distinct small molecules have been identified, which can bind to the SARS-CoV −1 PRF pseudoknot and disrupt its function [38–40]. In 2014, Ritchie et al. observed that the introduction of the small molecular ligand 2-{[4-(2-methylthiazol-4-ylmethyl)-[1,4]diazepane-1-carbonyl]amino} benzoic acid ethyl ester (MTDB) effectively decreased alternate folding by binding to the SARS-CoV −1 PRF pseudoknot [39]. The result is attributed to the possible hydrogen bonds formation of MTDB with the nucleotides in Loop 3, inhibiting a non-native folding path resulting in reduction of the −1 PRF rate to near 0%. Additional experiments find certain small molecules to be effective at inhibiting frameshifting for multiple coronaviruses–in both human and bat–showcasing the potential for pan-coronaviral therapeutic treatment design [41]. *In vitro* experiments for SARS-CoV-2 determined that MTDB binding reduced −1 PRF by 60% [9]. This result was replicated and expanded upon by experiments that found that various mutations in the SARS-CoV-2 −1 PRF pseudoknot sequence did not have a significant effect on anti-frameshifting activity of MTDB [25]. More recently, the compound merafloxacin was identified as significantly decreasing the efficiency of the −1 frameshift in SARS-CoV-2 [40]. This result was further validated by *in vivo* experiments [32], confirming the viability of targeting the frameshift element as a high-efficacy treatment. Another compound, 2-(5-acetylthiophen-2yl)furo[2,3-b]quinoline (KCB261770), was found to reduce frameshift efficiency in SARS-CoV, SARS-CoV-2, and MERS-CoV [42]. This points to possible structural similarity between the three coronaviruses, and makes KCB261770 a promising potential therapeutic for a range of coronaviruses. Increasing comprehensive structural knowledge of −1 PRF pseudoknots in coronaviruses can therefore be valuable in enhancing small molecule therapeutics.

SHAPE-MaP [43] (selective 2'-hydroxyl acylation analyzed by primer extension and mutational profiling) combines chemical probing with an energy model to identify structural motifs on entire viral RNA genomes [44]. High-throughput structure probing methods including

SHAPE-MaP and DMS-MaPseq [45] (dimethyl sulfate (DMS) mutational profiling with sequencing) measure many individual molecules and average the results to generate single-base resolution reactivity data. Recent studies performed *in vivo* SHAPE-MaP analyses of SARS-CoV-2 [12, 13, 46]. Despite improvements, high-throughput methods targeting the entire $\sim 30,000$nt genome are prone to noise and may suffer inaccuracies when predicting the structure for a specific region less than 100nt long. To focus on a smaller genomic region it can be beneficial to analyze predictions guided by *in vitro* SHAPE data on specific fragments of RNA, for example the SARS-CoV-2 frameshifting pseudoknot sequence [15].

In this work we present *Shapify*, an algorithm for prediction of RNA pseudoknotted structure based on the hierarchical folding hypothesis while incorporating reactivity data, to provide new insight to the SARS-CoV-2 −1 PRF pseudoknot structure. We utilized four structural reactivity probing experimental datasets [12, 13, 15, 46] as constraints to Shapify and compared predicted structures with those obtained using ShapeKnots [44], an existing heuristic algorithm for predicting RNA pseudoknotted structure and incorporating reactivity data. We demonstrate that Shapify improves the identification of probable structure formation paths for the SARS-CoV-2 −1 PRF pseudoknot.

In an attempt to expand knowledge of the SARS-CoV-2 −1 PRF pseudoknot structural conformation, in this work we first evaluate and report structural similarities between −1 PRF pseudoknots of SARS-CoV, SARS-CoV-2, and MERS-CoV. To further explore the hypothesized similarity between the SARS-CoV-2 and MERS-CoV −1 PRF pseudoknot structures [24], we enumerate the specific loci of similarity as well as the consensus structure. Following the hierarchical folding hypothesis (see Materials and methods), we predict structures for SARS-CoV-2 and MERS-CoV −1 PRF pseudoknots. To assess effects of mutations on the frameshifting structure, we provide predictions for seven SARS-CoV-2 −1 PRF mutated sequences and one MERS-CoV −1 PRF mutated sequence. These mutations were selected because they were observed in the population [28, 29] as well as experimentally validated for their effect on −1 PRF frameshifting [25].

Our results contribute to RNA structural prediction by providing a sampling of the landscape of notable–and previously unidentified–non-native secondary structures, which may play a role in regulating frameshifting [6, 47]. Whereas previous work recognized the importance of non-native pseudoknotted structures, the structural paths delineated here disclose non-native structure formation from initial stable stems. We discuss the relationship between each set of predicted initial stems and pseudoknotted structures. This approach is unique in providing information about possible paths to the final pseudoknotted structure. By analyzing predictions based on structural reactivity data, we provide novel information regarding application of such methods in this specific context (see Discussion). Finally, we contextualize our predictions with available experimental results including crystallography and cryo-electron microscopy [15, 16, 48]. Our pseudoknot structural predictions represent alternate starting points, which can improve the accuracy of existing 3-D physics-based modeling [17–22].

## Materials and methods

We first provide a background on RNA structure prediction, and present our sequence data sources and availability. Next we introduce methods for structural similarity detection and secondary structure prediction. That is, we introduce our *Shapify* algorithm and its closest competitor, ShapeKnots. Lastly, we provide information on the source and availability of the SHAPE reactivity data used in this work and introduce our procedure for SHAPE data analysis.

## Background on RNA structure prediction

Computational methods to predict RNA structure identify base pairs that form when RNA molecules fold. RNA folding refers to the process by which RNA acquires its structure through interacting nucleotide bases (also referred to as *bases*). While prediction of RNA tertiary structure (or 3D structure) is ultimately desirable, prediction of the secondary structure (set of all base pairs) is easier and sheds light on the tertiary structure. (For a recent review of methods for RNA tertiary structure prediction, we refer to Li et al. [49].) The majority of existing computational methods for the prediction of RNA structure, therefore, focus on the prediction of secondary structure. We represent an RNA molecule by its sequence *S* of length *n*. An RNA sequence is made up of four bases: Adenine (A), Cytosine (C), Guanine (G), and Uracil (U). When an RNA structure forms, complementary bases pair together and form hydrogen bonds. 'A' pairs with 'U' and 'G' pairs with either 'C' or 'U'–referred to as *canonical base pairs*. We refer to bases by their position in *S* indexed from 1 to *n* from 5' (left) to 3' (right) end. A *base pair* is then defined as the pairing of two bases *i* and *j* where $1 \leq i < j \leq n$, and represented as *i.j*. Consecutive base pairs are referred to as a *stem*. We note that each base can pair with at most one other base (i.e., no base triplets are allowed).

We say base pairs *i.j* and *i′.j′* are *nested* if $1 \leq i < i' < j' < j \leq n$, and *disjoint* if $1 \leq i < j < i' < j' \leq n$. An RNA structure with only nested and disjoint base pairs is referred to as a *pseudoknot-free* structure. An RNA structure is considered *pseudoknotted* when at least two of its base pairs, *i.j* and *i′.j′* cross: $1 \leq i < i' < j < j' \leq n$, in which case both *i.j* and *i′.j′* are considered pseudoknotted base pairs. We note that, while pseudoknotted base pairs are sometimes considered as part of the tertiary structure, we consider them as part of the secondary structure.

RNA folding can be described sequentially, with the *initial* stems forming first (Fig 3, red lines) followed by additional stems (Fig 3, blue lines). This concept is based on the *hierarchical folding hypothesis*. This hypothesis posits that RNA first folds into a pseudoknot-free structure, that is, one without crossing base pairs. Then, additional bases pair that may form pseudoknots to lower the minimum free energy of the structure [50]. The initial structure may be modified locally to accommodate formation of more stable base pairs [51, 52]. Hierarchical folding paths were experimentally identified in multiple pseudoknots [53], including frameshifting pseudoknots [54].

Assuming that the most stable RNA structure is the one with the lowest free energy, computational methods for prediction of RNA pseudoknotted structure based on the hierarchical folding hypothesis [55, 56], find the minimum free energy structure for a given sequence *S*, and a pseudoknot-free input structure *G*, where each RNA loop (i.e., an unpaired region of the RNA closed by a base pair) is assigned an energy value. The free energy of an RNA structure is

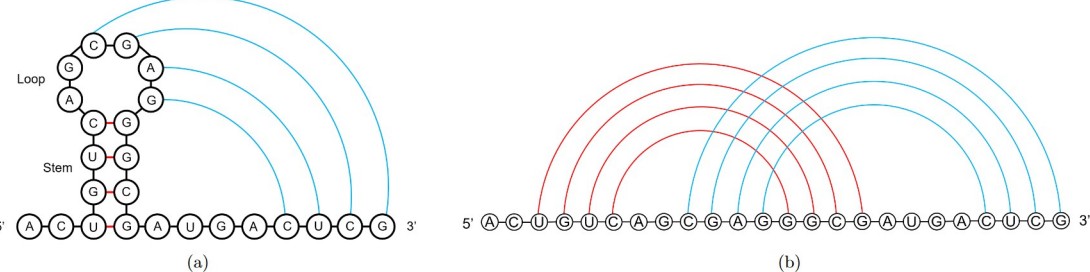

**Fig 3. Pseudoknot diagram.** A: Hairpin loop with stem base pairs shown in red. H-type pseudoknot pairing shown in blue between bases in the loop and complementary downstream bases. B: Representation of 3A in which arcs represent base pairs (arc diagram) for display of overlapping structures.

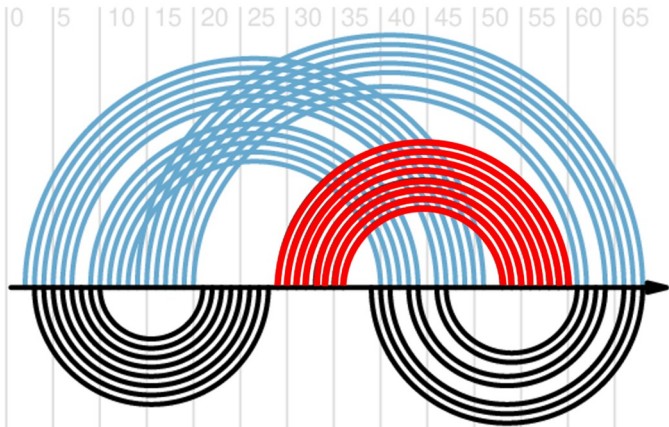

**Fig 4. Initial stem 2 suboptimal structure.** Predicted structures for the SARS-CoV-2 −1 PRF stimulating pseudoknot with initial Stem 2 (in red) as constraint. Blue arcs correspond with the minimum free energy structure, black arcs represent a suboptimal structure with only slightly higher free energy. Visualization generated using R-chie web server [71].

calculated as the sum of the energies of its loops. Free energy of some loops were experimentally determined, and others were extrapolated based on the experimental values [57–59].

While the HFold [55] algorithm adheres strictly to the hierarchical folding hypothesis, the Iterative HFold [56] algorithm allows for minimal modification in the input structure to accommodate formation of base pairs that can lower the energy of the structure. Iterative HFold utilizes four biologically sound methods and outputs the minimum free energy structure among these four predictions as the final structure. These four distinct methods explore different local structural modifications and allow for identification of *suboptimal* structures–structures with free energy only slightly higher than that of the minimum free energy structure. Here, we utilize all of these four structures (as opposed to just the one with the minimum free energy) to obtain a glimpse of the suboptimal structure landscape (cf. Fig 4).

## Data

We obtained the reference genomes for SARS-CoV, SARS-CoV-2, and MERS-CoV from the National Center for Biotechnology Information (NCBI) [28]. The position of frameshifting structure for each viral sequence was obtained from [9], and [24] as presented in Table 1.

The datasets generated and/or analysed during the current study are available in the frameshifting repository: github.com/HosnaJabbari/frameshifting.

## Structural similarity detection

We aligned whole genomes of SARS-CoV-2, SARS-CoV, and MERS-CoV using Clustal Omega [60] version 1.2.4, with SARS-CoV-2 as the reference. To detect *de novo* structural

**Table 1. List of viral sequences with their NCBI accession ID, position of frameshifting structure, length, and their reference.**

| Name | ID | Position | Length | Reference |
|---|---|---|---|---|
| SARS-CoV | NC_004718.3 | 13405–13472 | 68 | [9] |
| SARS-CoV-2 | NC_045512.2 | 13475–13542 | 68 | [9] |
| MERS-CoV | NC_019843.3 | 13440–13510 | 71 | [24] |

similarities in non-coding regions, we used RNAz v2.0 [61]. RNAz uses evolutionary conservation of functional secondary structures as well as thermodynamic stability of the secondary structure in detecting structural signal. We ran RNAz with our multiple sequence alignment of three sequences and false positive rate of about 1% (setting parameter $P > 0.9$; default value is 0.5).

## Secondary structure prediction

Following the HotSpots package of HotKnots V2.0 [59], we identified up to 20 most stable unique stems for each sequence. The stems were ranked based on their free energy and referred to by their IDs. These stems (referred to as *initial stems*) were used as constraints in Iterative HFold, Shapify, and ShapeKnots to explore the suboptimal structural landscape (cf. Fig A in S1 File: Hierarchical Folding Pipeline). The secondary structures produced based on each initial stem constraint were ranked by their free energy. Each predicted secondary structure was linked to the initial stem(s) that produced it. We report initial stems and the resulting pseudoknotted structures for eight SARS-CoV-2 −1 PRF stimulating pseudoknot sequences: reference, and seven mutated sequences. Additionally, we report initial stems and the resulting pseudoknotted structures for two MERS-CoV −1 PRF stimulating pseudoknot sequences: reference, and one mutated sequence.

## Shapify

In order to incorporate SHAPE reactivity data to guide the hierarchical folding approach, we adapted Iterative HFold [56] to develop Shapify. Shapify takes as input an RNA sequence, a SHAPE dataset, and a pseudoknot-free secondary structure (cf. Fig B in S1 File: Shapify Hierarchical Folding Pipeline) and outputs the predicted secondary structure. Both SHAPE data and the pseudoknot-free input structure guide Shapify's prediction with known RNA structural information. We used the pseudo energy terms created by Deigan et al. [62] from SHAPE reactivity data, as a means of integrating such data into our prediction algorithm. This pseudo energy term at index *i* of an RNA sequence includes a penalty for base pairing that increases with experimentally-derived SHAPE reactivity, *m*, (also referred to as *slope*) and an intercept that encourages base pairing for nucleotides with low SHAPE reactivity, *b*:

$$m[\log(SHAPE(i) + 1)] + b, \tag{1}$$

and is applied to stem energies only. Here, $SHAPE(i)$ refers to the SHAPE reactivity score at position *i* of the sequence.

The slope and intercept parameters must be determined empirically [44] for each prediction method, thus we aggregated a database of 30 RNAs with known structure and available SHAPE datasets. Data from the original ShapeKnots cross-validation [44] included five RNAs with lengths >300 nt, five riboswitch RNAs, four RNAs with structures that are not well predicted by thermodynamic parameters, and three RNAs whose structures are likely modulated by protein interaction. We supplemented this with six RNAs with known structure and available SHAPE data via the RNA Mapping Database [63]: three riboregulators involved with translation [64], SARS-CoV-2 3' and 5' UTR regions [65], and a ribonuclease domain of *Bacillus subtilis* [66].

We implemented a leave-one-out cross validation to determine the optimal values for slope *m* and intercept *b*, searching over a grid of 29 possible slope values and 21 possible intercept values (cf. Fig 5). For each combination we determined the geometric mean of the sensitivity and positive predictive value (ppv) for each of the RNA. We then averaged all the values excluding one RNA respectively, repeating the procedure for each possible RNA to be

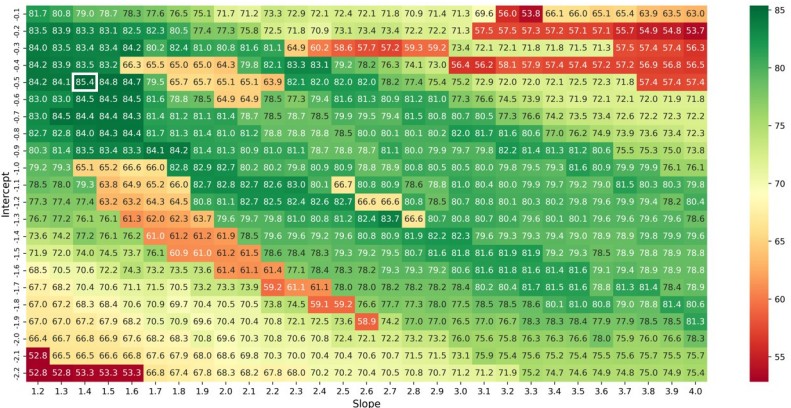

**Fig 5. Cross validation results.** For each possible combination of the intercept and slope, the value within the grid represents the geometric mean of the sensitivity and positive predictive value over 30 values of results from 29 RNA (each of the 30 RNA left out exactly once). Final result is the average of averages using a leave-one out scheme to avoid bias toward any one RNA. Optimal parameters identified by white box: intercept −0.5, slope 1.4. Color indicates performance relative to optimal with darker green as the best and red as the worst performance.

excluded and taking an average of averages to calculate the final result. Note that base pairs are considered to be predicted correctly even when one of the two indices is different by up to one nucleotide; this helps account for uncertainty and dynamism in RNA structure [67]. The longest RNA sequence of length 530 nt took 25 minutes for Shapify to deliver a prediction utilizing a 2.1GHz processor with 4GB memory. We determined the optimal values for Shapify parameters as slope $m = 1.4$ and intercept $b = −0.5$ (cf. Fig 5). Shapify is available at github.com/ltrinity/Shapify (DOI: 10.5281/zenodo.6100185).

## ShapeKnots

ShapeKnots [44] is a heuristic method similar to that of Ren et al. [68] incorporating SHAPE reactivity data for prediction of pseudoknotted structures. ShapeKnots utilizes the pseudo energy terms created by Deigan et al. [62]. ShapeKnots parameters were adjusted from default as follows: maximum number of internally generated structures = 200 (default 100), maximum percent difference in folding free energy change for internally generated suboptimal structures = 100% (default = 20%), maximum number of structures provided to user = 100 (default 20), maximum percent difference in folding free energy change for generating suboptimal structures = 50 (default 20). We compared performance of ShapeKnots and Shapify to provide a baseline.

## SARS-CoV-2 SHAPE data

We obtained four available SARS-CoV-2 SHAPE datasets as presented in Table 2.

**Table 2. List of SARS-CoV-2 SHAPE reactivity datasets used in this work with their referred name, type and reference.**

| Name | Type | Genome-wide | Reference |
|---|---|---|---|
| Huston et al. | *in vivo* | Yes | [12] |
| Manfredonia et al. | *in vivo* | Yes | [13] |
| Yang et al. | *in vivo* | Yes | [46] |
| Zhang et al. | *in vitro* | No | [15] |

Huston et al., Manfredonia et al., and Yang et al. are *in vivo* genome-wide probing, while Zhang et al. is *in vitro* probing specific to the SARS-CoV-2 frameshifting pseudoknot sequence. For each dataset, the values corresponding with the 68nt SARS-CoV-2 −1 PRF sequence were used as constraints in RNA secondary structure prediction. For both Shapify and ShapeKnots, in addition to the SHAPE reactivity information, a set of initial stems were provided as input to the algorithm.

### Bootstrapping

To further explore the degree to which the information in each SHAPE dataset was used in the structure prediction of Shapify and ShapeKnots, we implemented a bootstrap method as follows–for each original dataset, respectively. For each of the 68 indices of the SARS-CoV-2 −1 PRF sequence, the SHAPE reactivity value was selected randomly from the SHAPE dataset, with replacement. This newly generated bootstrap SHAPE information was then used as constraint for secondary structure prediction. This process was repeated 10,000 times for each of the four SHAPE datasets, for each of our prediction methods (i.e. Shapify and ShapeKnots). Having predicted 10,000 secondary structures for the bootstrap datasets, the secondary structures were converted into binary information–with 0 representing unpaired and 1 representing paired–and averaged for each position of the SARS-CoV-2 −1 PRF sequence.

### Results

We first describe the identified structural similarity among SARS-CoV, SARS-CoV-2, and MERS-CoV. Next, we present the predicted secondary structures of the SARS-CoV-2 and MERS-CoV −1 PRF pseudoknot using Iterative HFold and following the hierarchical folding hypothesis. Then we explore the effect of the single nucleotide mutations as stated in Materials and methods for SARS-CoV-2 and MERS-CoV. Finally, we visualize SHAPE-based predictions of the SARS-CoV-2 −1 PRF pseudoknot using Shapify, comparing them to ShapeKnots predictions as well as their dependence on the SHAPE data.

### −1 PRF structural similarity

One predicted loci of the structural similarity output from RNAz encompassed the region containing the −1 PRF pseudoknot for SARS-CoV-2. Structural similarity between SARS-CoV-2, SARS-CoV, and MERS-CoV was detected at locations 13439 to 13555, 13369 to 13485, and 13404 to 13523 in their respective aligned genomes (cf. Table 3). Structures are represented in dot-bracket format. Open parentheses show the base on the 5' side of the sequence and the closed parentheses represent the base on the 3' side of the sequence that are binding together. Each period "." (dot) identifies an unpaired base in the structure.

The alignment produced using RNAz identifies a highly conserved region for the −1 PRF structure of SARS-CoV-2 and MERS-CoV. This is interesting because there is not enough sequence similarity between the SARS-CoV-2 and MERS-CoV −1 PRF sequences to be detected using the Basic Local Alignment Search Tool [69]. For example, Lu et al. found SARS-CoV-2 genome to be ∼ 50% homologous to MERS-CoV, as compared to ∼ 79% homologous to SARS-CoV [70].

We note that while RNAz's prediction identifies conserved similarity in structures of SARS-CoV-2 and MERS-CoV, it only predicts pseudoknot-free structures (cf. Table 3), necessitating our further study of pseudoknots with minimum free energy methods.

**Table 3. −1 PRF Structural similarity for SARS-CoV, SARS-CoV-2, and MERS-CoV as predicted by RNAz.** Gaps in the alignment are represented as hyphen. Asterisks (*) in bottom row (PK location) correspond to the known location of SARS-CoV-2 and MERS-CoV −1 PRF native structures.

| Virus | −1 PRF Predicted Secondary Structure |
|---|---|
| SARS-CoV | ((((((((..(((((((((.(((((.....))))).....(((((((((((..((.--−−....))))))))))) (((((.......)))).)))).))))..))))).....))))) ............ |
| SARS-CoV-2 | .(((((((((....(((...(((((((.....))))).....))) (((((((((((..((.--−−....))))))))) (((((.....))) ..)))))))) ....)))) ..))))) |
| MERS-CoV | .....(((((((((...(((......((..............(((((((((((((((.....)))).)))))))))).. ((((((.....))))).....)))))))))) ((((........))) |
| Consensus | AGUCAGCUGAUGCAUCAACGGUUUUUAAAACGGGUGUAAGUGCA−−−GCCCGUCUUACACCGUGCGGCACAGGCACUAGUACUGAUGUCGUCUACAGGCGCUUUUGACAUCUAC |
| | .(((((....((((.....))))) ((((((((.((...)))))))))))..)))) ....((((.................)))))))) |
| PK location | − − − − − − − − − − − − − − − − − − − ******************************************--- − − − − − − − |

## SARS-CoV-2 −1 PRF pseudoknot

To predict the structure of the SARS-CoV-2 −1 PRF pseudoknot, following the hierarchical folding hypothesis we first generated a set of initial stems based on the reference sequence (see Materials and methods). Table 4 presents most stable initial stems for the SARS-CoV-2 −1 PRF stimulating pseudoknot ranked based on their free energies.

Using these 18 distinct initial stems as input constraints, we predicted 16 unique secondary structures for the SARS-CoV-2 −1 PRF pseudoknot reference sequence using Iterative HFold [56]. The top 11 most stable structures (based on their free energy) are presented in Table 5, see Table C in S1 File for the complete list. We use parentheses "()" and square brackets "[]" to represent crossing base pairs that identify pseudoknotted structures. The first column in Table 5 lists the initial stem ID(s) corresponding to the constraint that resulted in the predicted secondary structure. Note that the same structure can result from different initial stems because Iterative HFold allows for minor modifications to the input constraint.

The secondary structure with the lowest free energy as predicted by Iterative HFold is 100% consistent with the native SARS-CoV-2 −1 PRF pseudoknot structure (cf. Table 5, row 1). While some structures were native-adjacent–structurally close to the native structure–we also predicted native non-adjacent (markedly different) structures. In the third row of Table 5, for example, we have a structure with free energy of −18.26 kcal/mol that has the pseudoknot forming further towards the 3' end of the RNA compared to the native structure. Native non-adjacent pseudoknotted structures can be seen (cf. Table 5, rows 3, 8, 9 and 11) where native Stem 3 (cf. blue stem in Fig 2) does not form and expected unpaired bases in the second stem loop are paired.

**Table 4. The most stable initial stems in SARS-CoV-2 −1 PRF stimulating pseudoknot reference sequence, ranked based on their free energies.** These stems were used as structural constraint for predicting the SARS-CoV-2 −1 PRF stimulating pseudoknot secondary structure following the hierarchical folding hypothesis. First column provides stem ID (i.e. rank) and the third column lists free energy of the stem. Input sequence is provided in the bottom row.

| ID | Initial Stem | Free Energy (kcal/mol) |
|---|---|---|
| 1 | ( ( ( ( ( ( ( ( ( ( ( . . . . . . . . . . ) ) ) ) ) ) ) ) ) ) ) . . . . . . . . . . . . . . . . . . . . . . . . . . . . . . . . . . . . . . . . . . . | -10.79 |
| 2 | . . . . . . . . . . . . . . . . . . . . . . . . . . . ( ( ( ( ( ( ( ( . . . . . . . . . . . . . . . ) ) ) ) ) ) ) ) . . . . . . . . | -4.67 |
| 3 | . . . . . . . . . . . . . ( ( ( ( ( . . . . . . . . . . . . . . . . . . . . . . . . . . . . . . . . . . . . ) ) ) ) ) | -3.77 |
| 4 | . . . . . . . . . . . . . . . . . . . . . . . . . . ( ( ( ( . . . . . . . . ) ) ) ) . . . . . . . . . . . . . . . . . . . | -3.47 |
| 5 | . . . . . . . . ( ( ( ( ( . . . . . . . . . . . . . . . . . . . . . . ) ) ) ) ) . . . . . . . . . . . . . . . . . . . | -2.74 |
| 6 | . . . . . . . . . . . . . . . . . . . . . . . . . . . . . . . . . . . . . . . . ( ( ( ( . . . . . . . . . . . ) ) ) ) . . . . . . . . . | -2.54 |
| 7 | . . . . . . . . . ( ( ( ( . . . . . . . . . . . . . . . . . . . . . ) ) ) ) . . . . . . . . . . . . . . . . . . . . | -2.53 |
| 8 | . . . . . . . . . . . . . . . . ( ( ( . . . . . . . . . . . . . ) ) ) . . . . . . . . . . . . . . . . . . . . . . . | -2.42 |
| 9 | . . . . . . . . . . . . . . . . . ( ( ( ( . . . . . . . . . . . . ) ) ) ) . . . . . . . . . . . . . . . . . . . | -2.35 |
| 10 | ( ( ( ( . . . . . . . . . . . . . ) ) ) ) . . . . . . . . . . . . . . . . . . . . . . . . . . . . . . . . . . . . . | -2.26 |
| 11 | . . . . . . . . . . . . . . . ( ( ( . . . . . . . . . . . . . . . . . . . ) ) ) . . . . . . . . . . . . . . . . . . | -2.10 |
| 12 | . . . . . . . . . . . . . . . . . . . ( ( ( ( . . . . . . . . . . . . . ) ) ) ) . . . . . . . . . . . . . . . . . | -2.07 |
| 13 | . . . . . . . . ( ( ( ( ( . . . . . . . . . . . . . . . . . . . . . . . . . ) ) ) ) ) . . . . . . . . . . . . . . | -1.36 |
| 14 | . . . . . . . . . . . ( ( ( ( . . . . . . . . . . . . . . ) ) ) ) . . . . . . . . . . . . . . . . . . . . . . | -1.32 |
| 15 | . . ( ( ( ( ( . . . . . . . . . . . . . . . . . . . . . . . . ) ) ) ) ) . . . . . . . . . . . . . . . . . . . | -0.66 |
| 16 | . . . . . . . . . . . . . . . . . . . . . ( ( ( ( ( . . . . . . . . . . . . . . . . . . . . . . . . . . . . . . ) ) ) ) ) . | -0.51 |
| 17 | . . . ( ( ( ( . . . . . . . . . . . . . . . . . . . . . . . . . . . ) ) ) ) . . . . . . . . . . . . . . . . . . | -0.38 |
| 18 | ( ( ( ( ( ( ( . . . . . . . . . . . . . . . . . . . . . . . . . . . . . ) ) ) ) ) ) ) . . . . . . . . . . . | -0.24 |
| Sequence | GCGGUGUAAGUGCAGCCCGUCUUACACCGUGCGGCACAGGCACUAGUACUGAUGUCGUAUACAGGGCU | |

**Table 5. Predicted secondary structures for the SARS-CoV-2 −1 PRF stimulating pseudoknot based on the reference sequence.** These structures are predicted by Iterative HFold given the initial stems in Table 4 as structural constraints. Certain suboptimal structures (e.g. $6_s$ and $2_s$, cf. Fig 4) are reported, because in these cases the structure has only slightly higher free energy than the minimum free energy structure predicted by Iterative HFold. Native structure is marked with an asterisk (*) in row 1. As shown in rows 1, 2 and 5 of the table, multiple initial stems can result in a single prediction for the −1 PRF stimulating pseudoknot. Input sequence is provided in the bottom row.

| Initial Stem ID | Predicted Secondary Structure | Free Energy (kcal/mol) |
|---|---|---|
| 1, 3, 9, 18 | ( ( ( ( ( ( ( ( ( ( ( . . . [ [ [ [ [ [ [ ) ) ) ) ) ) ) ) ) ) ) ( ( ( ( ( ( ( ( ( . . . . . . . . . ) ) ) . ) ) ) ) ) ) ) . . ] ] . ] ] ] ] ] * | -18.86 |
| 8, 11 | ( ( ( ( ( ( ( ( ( ( ( . ( ( [ [ [ [ [ ) ) ) ) ) ) ) ) ) ) ) ) ( ( ( ( ( ( ( ( ( . . . . . . . . . ) ) ) . ) ) ) ) ) ) ) . . . . . ] ] ] ] ] | -18.80 |
| 4 | ( ( ( ( ( ( ( ( ( ( ( . . . . . . . . . . ) ) ) ) ) ) ) ) ) ) ) . . ( ( ( ( ( ( ( ( [ [ . [ [ . [ [ [ ) ) . ) ) ) ) . . . ] ] ] ] ] . ] ] . | -18.26 |
| 14 | . ( ( ( ( ( ( ( ( ( ( ( . . . [ [ [ [ [ [ [ ) ) ) ) ) ) ) ) ) ) ) ( ( ( ( ( ( ( ( ( . . . . . . . . . ) ) ) . ) ) ) ) ) ) ) ) . ] ] . ] ] ] ] ] | -17.82 |
| 5, 13, 15 | . . ( ( ( ( ( ( ( ( ( ( . . . [ [ [ [ [ [ [ ) ) ) ) ) ) ) ) ) ) ) ( ( ( ( ( ( ( ( ( . . . . . . . . ) ) ) . ) ) ) ) ) ) ) ) ) ] ] . ] ] ] ] ] | -17.63 |
| 7 | ( ( ( ( ( ( ( ( ( ( . . . . [ [ [ [ [ [ [ . ) ) ) ) ) ) ) ) ) ) ) ( ( ( ( ( ( ( ( ( . . . . . . . . . ) ) ) . ) ) ) ) ) ) ) . . ] ] . ] ] ] ] ] | -17.37 |
| 12 | ( ( ( ( ( ( ( ( ( . . . . . [ [ [ [ [ [ [ . . ) ) ) ) ) ) ) ) ) ( ( ( ( ( ( ( ( ( . . . . . . . . . ) ) ) . ) ) ) ) ) ) ) . . ] ] . ] ] ] ] ] | -16.62 |
| 6 | ( ( ( ( ( ( ( ( ( ( ( . . . . . . . . . ) ) ) ) ) ) ) ) ) ) ) ( ( ( ( ( ( . . . . . . . . [ [ [ . [ [ [ . ) ) ) ) ) ) . . . ] ] ] . ] ] ] | -16.24 |
| $6_s$ | ( ( ( ( ( ( ( ( ( ( ( . . . . . . . . . . ) ) ) ) ) ) ) ) ) ) ) ) . . . . . . . . ( ( ( ( . . [ [ [ . [ [ [ . ) ) ) ) . . . . . ] ] ] . ] ] ] | -16.08 |
| 2 | . ( ( ( ( ( ( ( . ( ( ( ( ( [ [ [ [ [ [ [ . . . . . . . . [ [ [ [ [ [ [ . . . ) ) ) ) ) . ) ) ) ) ) ) ) . ] ] ] ] ] ] ] ] ] ] ] ] . ] ] ] ] ] | -15.26 |
| $2_s$ | . . ( ( ( ( ( ( ( ( . . . . . . . . . ) ) ) ) ) ) ) ) ) ( ( ( ( ( ( ( ( . . [ [ [ . [ [ . [ [ . . . . . ) ) ) ) ) ) ) ) ) ] ] ] . ] ] ] | -14.84 |
| Sequence | GCGGUGUAAGUGCAGCCCGUCUUACACCGUGCGGCACAGGCACUAGUACUGAUGUCGUAUACAGGGCU | |

## MERS-CoV −1 PRF pseudoknot

Table 6 presents initial stems for the MERS-CoV −1 PRF stimulating pseudoknot, as predicted for the reference sequence. For the complete set of stems see Table D in S1 File. Using the 18 initial stems as input constraints, 15 unique secondary structures were predicted for the MERS-CoV −1 PRF pseudoknot reference sequence using Iterative HFold. These predicted pseudoknotted structures are presented in Table 7 sorted by their free energy, see Table E in S1 File for the complete list. Note that multiple structures were reached from more than one starting point (cf. Table 7, rows 1 & 2). There is no well established native structure for the MERS-CoV −1 PRF pseudoknot. The secondary structure with the lowest free energy as predicted by Iterative HFold is 93% consistent with the structure presented by Fourmy and Yoshizawa [24]. Comparing the minimum free energy structure predicted for MERS-CoV and SARS-CoV-2 −1 PRF pseudoknots (cf. Tables 5 and 7, row 1), we see the familiar three stems and H-type pseudoknotted structure. In the MERS-CoV −1 PRF pseudoknot predictions, the second and third lowest free energy structures are pseudoknot-free. Furthermore, only six of the 15 predicted secondary structures were pseudoknotted. By contrast, each of the 16 structures predicted for the SARS-CoV-2 −1 PRF pseudoknot (cf.

**Table 6. Top four energetically favourable stems predicted for MERS-CoV −1 PRF stimulating pseudoknot reference sequence.** These stems were used as structural constraint to Iterative HFold program to predict the secondary structure of the −1 PRF stimulating pseudoknot. First column provides stem IDs, the second column presents the predicted stems in dot-bracket format and the third column lists free energy of the given stem in kcal/mol. Input sequence provided in bottom row. See Table D in S1 File for complete table.

| ID | Stem | Free Energy (kcal/mol) |
|---|---|---|
| 1 | ( ( ( ( ( ( ( ( ( ( ( ( . . . . . . . . . . . ) ) ) ) ) ) ) ) ) ) ) ) . . . . . . . . . . . . . . . . . . . . . . . . . . . . . . . . . . . . . . . . | -10.39 |
| 2 | . . . . . . . . . . . . . . . . . . . . . . . . . . . . . . . . . . . ( ( ( ( ( . . . . . ) ) ) ) ) . . . . . . . . . . . . . . . . . . . . | -4.69 |
| 3 | . ( ( ( ( . . . . . . . . . . . . ) ) ) ) ) . . . . . . . . . . . . . . . . . . . . . . . . . . . . . . . . . . . . . . . . . . . . . . . . . . | -4.25 |
| 4 | . . . . . . . . . . . . . . . ( ( ( ( ( . . . . . . . . . . . . . . . . . . . . . . . . . . . . . . . . . . . . . . . . ) ) ) ) ) | -3.79 |
| Input sequence | GGGGUUCUAUUGUAAAUGCCCGAAUAGAACCCUGUUCAAGUGGUUUGUCCACUGAUGUCGUCUUUAGGGCA | |

**Table 7. Top four lowest free energy secondary structures predicted for MERS-CoV −1 PRF stimulating pseudoknot based on the reference sequence.** These structures are predicted by Iterative HFold given the input stems in Table 6 as structural constraints. Input sequence is provided in the bottom row. See Table E in S1 File for the complete table.

| Initial Stem ID | Secondary Structure Prediction | Free Energy (kcal/mol) |
|---|---|---|
| 1, 4 | ( ( ( ( ( ( ( ( ( ( ( ( . . . . . [ [ [ [ [ . ) ) ) ) ) ) ) ) ) ) ) ) . . ( ( ( . ( ( ( ( . . . . . ) ) ) ) ) ) ) . . . . . . . . . . . ] ] ] ] ] | -17.36 |
| 2, 3, 8 | ( ( ( ( ( ( ( ( ( ( ( . . . . . . . . . . . . ) ) ) ) ) ) ) ) ) ) ) . . . . . ( ( ( ( ( . . . . . ) ) ) ) ) . . . . . . ( ( ( . . . . . ) ) ) . | -16.40 |
| 12 | ( ( ( ( ( ( ( ( ( ( ( . . . . . . . . . . . . ) ) ) ) ) ) ) ) ) ) ) . . . . . ( ( ( ( ( . . . . . ) ) ) ) ) . . . . . . ( ( ( ( . . . ) ) ) ) . | -16.17 |
| 13 | ( ( ( ( ( ( ( ( ( ( ( . . . . . . . . . . . . ) ) ) ) ) ) ) ) ) ) ) . . . . . ( ( ( ( ( . . . . . ) ) ) ) ) . . ( ( ( ( . . . . . . . . ) ) ) ) | -16.14 |
| Input sequence | GGGGUUCUAUUGUAAAUGCCCGAAUAGAACCCUGUUCAAGUGGUUUGUCCACUGAUGUCGUCUUUAGGGCA | |

Table 5) contained pseudoknotted base pairs. The energy of predicted secondary structures for the MERS-CoV −1 PRF pseudoknot are generally higher compared to structures predicted for SARS-CoV-2, indicating they may be less stable. In addition, there is a bigger energy gap between the minimum free energy predicted structure for MERS-CoV and the second most energetically favourable structure (0.94 kcal/mol); as compared with SARS-CoV-2 where there are 3 structures within 0.94 kcal/mol of the minimum free energy structure.

## Effect of mutations on the SARS-CoV-2 −1 PRF pseudoknot

We repeated the hierarchical folding method for each mutated sequence, see red arrows in Fig 2 for mutation locations in the native structure. We note that Neupane et al. observed a significant decrease of frameshifting efficiency in SARS-CoV-2 −1 PRF for only U20C mutation [25] while Ishimaru et al. observed a decrease in frameshifting efficiency of SARS-CoV with C43U and U47C [35].

The predicted initial stems did change in some cases between the reference sequence and the mutated sequences. Certain stems that were predicted for both the reference sequence and respective mutated sequences had differences in free energy. Some stems were stable for the reference sequence but destabilized by mutated sequences. In addition, specific mutated sequences led to initial stems that were not identified for the reference sequence. We present novel initial stems for mutated sequences ranked by their free energies and referred to by their IDs based on where they would have been ranked by free energy relative to the initial stems of the reference sequence (cf. Table 8). Initial stems for mutated sequences are given an ID with a letter to distinguish them from the reference sequence stems (e.g., 5*a*). If multiple stems from mutated sequences have the same free energy ranking with respect to the initial stems of the reference sequence, they are given IDs with sequential letters (e.g., 5*a* and 5*b*).

The initial stems predicted for the C13U mutated sequence destabilized initial stems 5 and 7 (as their free energy increased). One novel stem was detected for this mutation, referred to as 13*a* (cf. Table 8).

The initial stems predicted for the U20C mutated sequence destabilized initial stem 1, but stabilized stems 9, 10, 12, and 16 (their free energies decreased). Two novel stems were detected for the U20C mutated sequence referred to as 5*a* and 5*b*.

Repeating the method for the sequence with the guanine/uracil mutation at position 29 (G29U) did not change the predicted energy of any of initial stems, but did not detect four stems (initial stems 1, 2, 4 and 14). Five novel stems were detected based on G29U mutation, referred to as 2*a*, 3*a*, 11*a*, 13*b*, and 15*a*.

**Table 8. Predicted initial stems for mutated SARS-CoV-2 −1 PRF stimulating pseudoknot sequences.** These stems were used as structural constraint for predicting the secondary structure of mutated SARS-CoV-2 −1 PRF sequences. First column identifies the mutation and its location in the 68 nt −1 PRF sequence. For example C13U identifies a mutation from C to U at index 13. Second column provides the stem ID based on the stem's free energy and the ranking of the initial stem of the reference sequence. For example, in row 1 the stem has a free energy of −1.87 kcal/mol and is denoted by stem ID 13*a*. Relative to the initial stems predicted for the reference sequence (cf. Table 4), this stem has the thirteenth lowest free energy. Third column represents the predicted initial stem for the mutated sequence, and the fourth column provides free energy of the given stem (kcal/mol). Input sequence is provided in the bottom row for each mutation section with mutations highlighted in yellow.

| Mutation | ID | Stem | Free Energy (kcal/mol) |
|---|---|---|---|
| C13U | 13a | . . . . . . . . . ( ( ( ( ( ( . . . . . . ) ) ) ) ) ) . . . . . . . . . . . . . . . . . . . . . . . . . . . . . . . . . . . . . . . | -1.87 |
| | | GCGGUGUAAGUG*U*AGCCCGUCUUACACCGUGCGGCACAGGCACUAGUACUGAUGUCGUAUACAGGGCU | |
| U20C | 5a | . . . . . . . . . . . . . . . . . ( ( ( . . . . . . . . . . . . . ) ) ) . . . . . . . . . . . . . . . . . . . . . . . . . . . . . . | -2.99 |
| | 5b | . . ( ( ( ( . . . . . . . . . . . . ) ) ) ) . . . . . . . . . . . . . . . . . . . . . . . . . . . . . . . . . . . . . . . . . . . . | -2.77 |
| | | GCGGUGUAAGUGCAGCCCG*C*CUUACACCGUGCGGCACAGGCACUAGUACUGAUGUCGUAUACAGGGCU | |
| G29U | 2a | . . ( ( ( ( ( ( ( ( . . . . . . . . . . ) ) ) ) ) ) ) ) . . . . . . . . . . . . . . . . . . . . . . . . . . . . . . . . . . . . . . | -8.39 |
| | 3a | . . . . . . . . . . . . . . . . . . . . . . . . . . . ( ( ( ( ( ( . . . . . . . . . . . . . . . . . . ) ) ) ) ) ) . . . . . . . . | -3.86 |
| | 11a | . . . . . . . . . . . . . . . . . . . . . . ( ( ( ( ( . . . . . . . . . . . . . . . . . . . . . . . . . . . ) ) ) ) ) . . | -2.13 |
| | 13b | . . . . ( ( ( ( ( ( . . . . . . . . . . . . . . . ) ) ) ) ) ) . . . . . . . . . . . . . . . . . . . . . . . . . . . . . . . . . . | -2.04 |
| | 15a | . . . . . . . . . . ( ( ( ( ( . . . . . . . . . . . ) ) ) ) ) . . . . . . . . . . . . . . . . . . . . . . . . . . . . . . . . . . | -0.92 |
| | | GCGGUGUAAGUGCAGCCCGUCUUACACC*U*UGCGGCACAGGCACUAGUACUGAUGUCGUAUACAGGGCU | |
| C43U | 4a | . . . . . . . . . . . . . . . . . . . . . . . . . . . . . . . . . ( ( ( ( ( ( ( . . . . ) ) ) ) ) ) ) . . . . . . . . . . . . . . | -3.74 |
| | | GCGGUGUAAGUGCAGCCCGUCUUACACCGUGCGGCACAGGCA*U*UAGUACUGAUGUCGUAUACAGGGCU | |
| U47C | 4b | . . . . . . . . . . . . . . . . . . . . . . . . . ( ( ( ( . . . . . . . . . . . . . . . ) ) ) ) . . . . . . . . . . . . . . . . . . | -3.54 |
| | 8a | . ( ( ( ( ( ( . . . . . . . . . . . . . . . . . . . . . . . . . . . . ) ) ) ) ) ) . . . . . . . . . . . . . . . . . . . . . . . . | -2.46 |
| | | GCGGUGUAAGUGCAGCCCGUCUUACACCGUGCGGCACAGGCACUAG*C*ACUGAUGUCGUAUACAGGGCU | |
| U58C | 18a | . . . . . . . . . ( ( ( ( . . . . . . . . . . . . . . . . . . . . . . . . . . . . . . . . . . . . . . . . ) ) ) ) . . . . . . . . . | -0.27 |
| | | GCGGUGUAAGUGCAGCCCGUCUUACACCGUGCGGCACAGGCACUAGUACUGAUGUCG*C*AUACAGGGCU | |
| C62U | 13c | . . . . . . . . . . . . . . . . . . ( ( ( ( ( ( . . . . . . . . . . . . . . . . . . . . . . . . . . . . . . . . . ) ) ) ) ) ) . | -1.96 |
| | | GCGGUGUAAGUGCAGCCCGUCUUACACCGUGCGGCACAGGCACUAGUACUGAUGUCGUAUA*C*AGGGCU | |

For the sequence with the cytosine/uracil mutation at position 43 (C43U), initial stems 4 and 5 were destabilized. Initial stems 6 and 15 were not detected for C43U, but a novel stem was detected, referred to as 4*a*.

Initial stems predicted for the sequence with the uracil/cytosine mutation at position 47 (U47C) stabilized initial stem 13. Two novel stems were detected for the U47C mutation, referred to as 4*b* and 8*a*.

For the sequence with the uracil/cytosine mutation at position 58 (U58C), initial stems 2 and 18 were stabilized. One novel stem was detected for the U58C mutation, referred to as 18*a*.

Finally, for the sequence with the cytosine/uracil mutation at position 62 (C62U), initial stem 16 was not detected. There was a novel stem identified for this mutation, referred to as 13*c*.

We used the 18 initial stems from the reference sequence in addition to the novel initial stems for respective mutated sequences (cf. Table 8) as input constraints to predict secondary structures using Iterative HFold. For each of the seven SARS-CoV-2 −1 PRF pseudoknot mutated sequences, 14 − 19 secondary structures were predicted (cf. Tables G-M in S1 File). The native structure, the structure with the minimum free energy in Table 5, was not predicted for U20C, G29U, or C62U mutated sequences based on any of the initial mutated stems. For example, initial stems 1 and 3 that would result in the native structure based on the reference sequence, did not result in the native structure based on the sequence with the U20C mutation (cf. Table H in S1 File). Instead, they joined initial stems 8 and 11 resulting in a non-native

structure. In general two to five structure clusters (identical prediction from multiple different initial stems) were identified for each mutated sequence.

## Effect of mutation on the MERS-CoV −1 PRF pseudoknot

We obtained 237 MERS-CoV genomes from the NCBI Virus Variation database [28]. There was a mutation observed at position 13479 from C to U in three sequences (KR011263, MG011354, and KR011266). Following the procedure explained in Materials and methods, we obtained initial stems for the mutated sequence: initial stem 12 was destabilized, and there was a novel initial stem detected (cf. Table D in S1 File). We used the 18 initial stems as input constraints to predict secondary structures using Iterative HFold leading to 13 unique secondary structure predictions (cf. Table N in S1 File). The minimum free energy structure predicted for reference and mutated sequence remained the same. The mutated base was only predicted as paired in one out of the 13 structures, although pre-mutation it was paired in 12 out of 15 of the structures predicted for the reference sequence. Notably, initial stems 2, 3, and 8 led to a different structure, accommodating the mutation with a larger loop.

## SARS-CoV-2 −1 PRF pseudoknot with SHAPE

In this section we present Shapify results using the 18 initial stems (cf. Table 4) and four SHAPE datasets (see Materials and methods) as constraint. We found significant overlap among the four sets of predictions, giving a total of 43 unique secondary structure predictions (cf. Table O in S1 File).

In Figs 6 and 7 we visualize structural paths from the initial stems to the predicted secondary structures from Shapify. Beginning from the left, each initial stem is labeled with its ID (cf. Table 4). There are four predictions obtained for each initial stem, one for each of the SHAPE datasets used. Note that predictions among the four results could be the same. Here we include any additional suboptimal structures within 2 kcal/mol of the minimum free

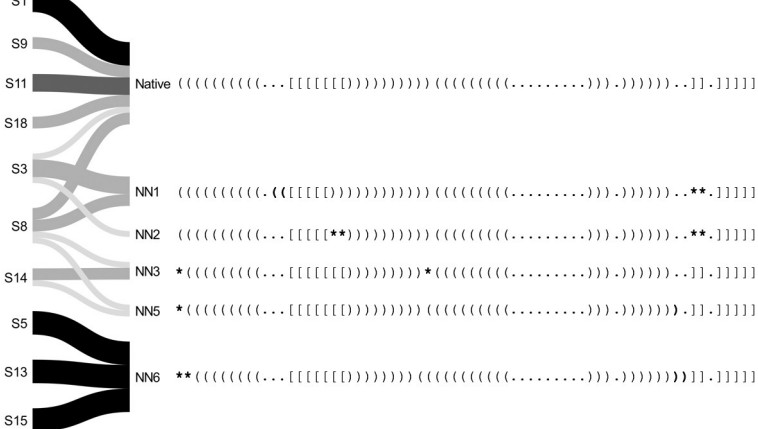

**Fig 6. Shapify predicted native-adjacent structural paths.** Presentation of structural paths from each initial stem leading to native or non-native (but native-adjacent) secondary structures (e.g., NN1 is the lowest free energy non-native structure; cf. Table O in S1 File). Initial stems labeled on the left (e.g., S1 for initial stem 1; cf. Table 4). If the structure predicted for a specific initial stem was the same for all four SHAPE datasets it is presented with a black colored path. In other cases, where the predicted structure was the same for three, two, or only one of the SHAPE datasets, the path is colored dark grey, grey, or light grey, respectively. Differences from the native structure are marked in bold, with parentheses/brackets representing changes in paired bases, and asterisks representing predicted unpaired bases that were paired in the native structure.

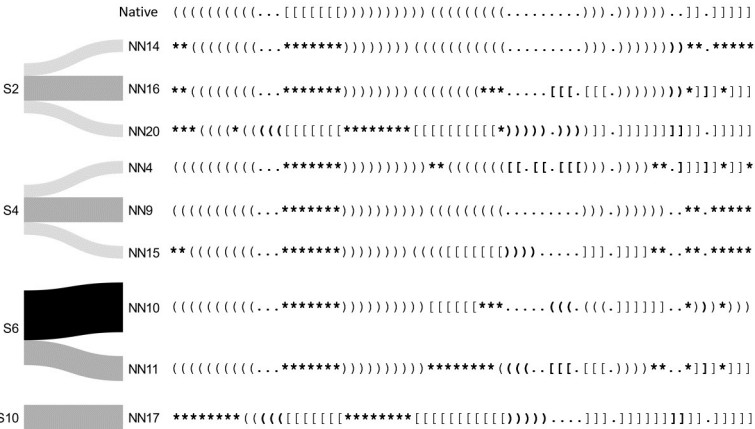

**Fig 7. Shapify predicted native non-adjacent structural paths.** Presentation of structural paths from the initial stems leading to native non-adjacent secondary structures. Initial stems are labeled on the left (cf. Table 4). If, for a given initial stem, the predicted structure is the same for all four SHAPE datasets, it is presented with a black colored path. In other cases, where the predicted structure is the same for three, two, or only one of the SHAPE datasets, the path is colored dark grey, grey, or light grey, respectively. Differences from native structure are marked in bold, with parentheses/brackets representing changes in paired bases, and asterisks representing predicted unpaired bases that are paired in the native structure.

energy prediction. Darker color represents higher agreement among SHAPE datasets, meaning the same prediction was obtained with different SHAPE data.

Figs 6 and 7 present structural paths from the initial stems to the native and non-native structures predicted by our Shapify algorithm. All non-native structures are numbered according to their free energy (e.g., NN1 is the lowest free energy non-native structure, NN2 is the second lowest free energy non-native). The initial stem IDs are annotated for each secondary structure and SHAPE dataset along with all structures' free energies in Table O in S1 File. For additional visualization with initial stems highlighted, cf. Fig C-D in S1 File. The native structure and five native-adjacent structures (non-native secondary structures that have only minor differences from the native structure) are included in Fig 6. Separately, we visualize nine native non-adjacent structures that are significantly different from the native structure (cf. Fig 7). Differences between the native structure in the first row and other structures below are indicated in bold; brackets show differences in paired bases, and asterisks identify bases that are paired in the native structure but predicted unpaired in the non-native structure.

The native structure was reached by six different initial stems (cf. Fig 6). This indicates that there are structural paths to the native structure from an array of initial stems. Native-adjacent structures, also display significant diversity in the paths with which they can be reached. Specifically, non-native structures 1, 3, 5, and 6 (i.e. the first, third, fifth, and sixth lowest free energy non-native structures) which are classified as native-adjacent were each reached by two different initial stems. For native non-adjacent structures, however, there were not multiple paths to the same structure from different initial stems. Furthermore, there was less agreement between predictions from different SHAPE datasets in structural paths to the native non-adjacent structures (presented with lighter path colors).

## ShapeKnots predictions

We obtained predictions for the SARS-CoV-2 −1 PRF pseudoknot using ShapeKnots and each of the SHAPE datasets. In addition, since ShapeKnots is capable of receiving input structures, we further execute ShapeKnots with each SHAPE dataset and initial stems (cf. Table 4),

**Table 9. Most energetically favourable ShapeKnots [44] secondary structure predictions for the SARS-CoV-2 −1 PRF stimulating pseudoknot based on the reference sequence.** SHAPE dataset source for each prediction indicated in the first column. Second column provides stem IDs, NA if none used. Certain suboptimal structures (e.g., $2_s$) are reported because of only slightly higher free energy than the minimum free energy structure predicted by ShapeKnots.

| SHAPE Data | Initial Stem ID | Secondary Structure Prediction | Free Energy (kcal/mol) |
|---|---|---|---|
| Huston et al. [12] | NA | .(((((((((.(([[[[[])))))))))))))((((((((((.........))).)))))))))....]]]]] | -18.11 |
| | 2 | ..(((((((((.(([[[[[])))))))))))))((((((((((.........))).)))))))))...]]]]] | -17.89 |
| Zhang et al. [15] | NA | .(((((((((...[[[[[..))))))))))))((((((((((.........))).)))))))))....]]]]] | -17.85 |
| Manfredonia et al. [13] | NA | .(((((((((...[[[[[[[])))))))))))((((((((((((....)).))))))))))))....]].]]]]] | -17.25 |
| Zhang et al. [15] | $2_s$ | [[(((((((((.]][[[[[..))))))))))))((((((((((.........))).)))))))))...]]]]] | -16.95 |
| Manfredonia et al. [13] | $2_s$ | [[(((((((((.]][[[[[[])))))))))))((((((((((((....)).))).))))))))))]].]]]]] | -16.38 |
| Zhang et al. [15] | 1 | (((((((((.........)))))))))))((((((((((.........))).)))))))).......... | -16.32 |
| Huston et al. [12] | $1_s$ | (((((((((.((.....)))))))))))))((((((((((.........))).))))))))........... | -15.13 |
| Yang et al. [46] | $1_s$ | | |
| Manfredonia et al. [13] | $1_s$ | (((((((((((.((.....)))))))))))))).(((((((((((....)).))).)))))............. | -14.57 |
| Sequence | | GCGGUGUAAGUGCAGCCCGUCUUACACCGUGCGGCACAGGCACUAGUACUGAUGUCGUAUACAGGGCU | |

respectively, as constraints. The lowest free energy structures predicted by ShapeKnots were obtained when no initial stem was used as input to the program (i.e. only RNA sequence and SHAPE data was used as input, cf. Table 9). As shown in Fig 8, structures predicted by Shape-Knots, have generally higher free energy compared to the structures predicted by Shapify.

Initial stems 1 and 2 as input constraints led to predictions with slightly higher free energy (compared to the ones predicted by Shapify), and each had multiple suboptimal structures. Interestingly, none of the structures predicted by ShapeKnots when given initial stem 1 as input constraint were pseudoknotted (cf. Table 9, bottom four rows). This is in contrast to Shapify's prediction that identifies initial stem 1 as a structural path to the native pseudoknotted structure with high agreement among all SHAPE datasets (cf. Fig 6). We note that in the absence of SHAPE reactivity data, Iterative HFold also identified the initial stem 1, as one of the initial stems that reach the native pseudoknotted structure (cf. Table 5, row 1).

ShapeKnots identified two paths from initial stem 2 leading to energetically favourable native-adjacent structures that contained multiple distinct sets of pseudoknotted base pairs (cf. Table 9, rows 5 and 6). These new pseudoknotted base pairs create a kissing-hairpin structure as opposed to the native H-type pseudoknotted structure. The initial stem 2 also has a path to reach an H-type pseudoknotted structure that resembles the native structure with a shift of Stem 1 to the 3' end.

## SARS-CoV-2 −1 PRF pseudoknot SHAPE data analysis

Fig 9A presents comparison of SHAPE reactivity values for each position in the 68nt −1 PRF SARS-CoV-2 sequence reported by Manfredonia et al. [13], Huston et al. [12], Yang et al. [46], and Zhang et al. [15]. A base is considered reactive (unpaired) if its reactivity score is above 0.3.

There are five regions that are identified as reactive by at least two SHAPE datasets: bases 6 − 13, 19 − 32, 41 − 48, 51 − 52, 59 − 63, 67 − 68. The loop region from positions 60 − 63 was well captured by the Zhang et al. and Yang et al. datasets, with Huston et al. and Manfredonia

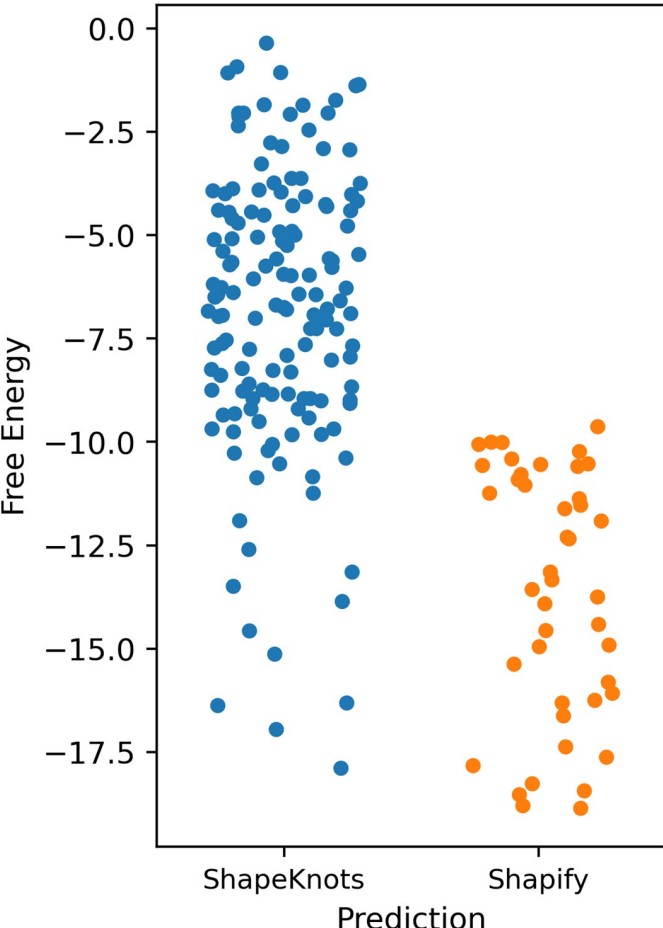

**Fig 8. Comparison of free energy of ShapeKnots and Shapify structural predictions.** Each dot represents a unique secondary structure prediction for SARS-CoV-2 −1 PRF frameshifting pseudoknot. The *y*-axis represents free energy in kcal/mol with lower free energy corresponding with more energetically favourable structures. ShapeKnots (blue) failed to predict the most energetically favourable structures when given SHAPE data and initial pseudoknot free stems as constraint as compared to Shapify (orange). All structures are listed in Tables O-P in S1 File.

et al. also finding reactivity in this region. There were limited positions that all SHAPE datasets agreed reported no reactivity above the threshold: positions $1 − 3$, $15 − 16$, 18, $33 − 34$, $36 − 37$, $49 − 50$, and 54. For significant reactivity, all datasets agreed only on positions 23 and 29.

Fig 9B presents ShapeKnots' predictions using bootstrapped SHAPE values repeated 10,000 times and averaged for each index (where 0 identifies an unpaired base and 1 a paired base). A mean values of 1.0 at index *i* conveys the nucleotide at position *i* was predicted as paired for all bootstrapped datasets, and mean value of 0.0 at position *i* indicates that nucleotide *i* was always predicted as unpaired for all bootstrapped datasets.

Fig 9C similarly presents Shapify's predictions using bootstrapped SHAPE values.

There were significant differences with respect to the predictions via ShapeKnots and Shapify following bootstrapping. Shapify predicted the first and second bases in the sequence to be paired far more frequently than ShapeKnots did. This is interesting because, despite general consensus on all SHAPE datasets for no reactivity for these positions (i.e. paired), three energetically favourable native-adjacent structures (NN3, NN5, and NN6, cf. Fig 6) were predicted to have these base(s) unpaired.

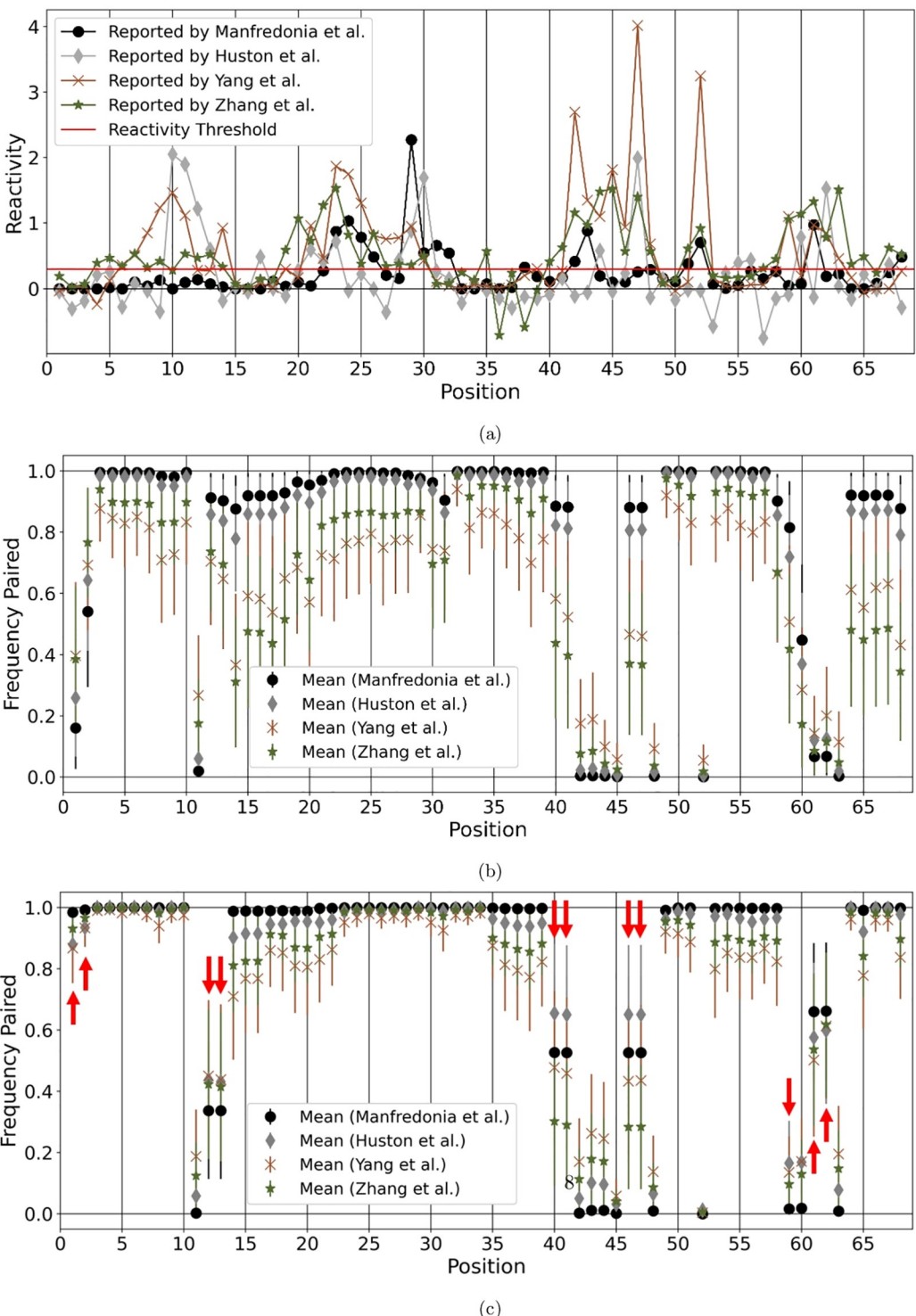

**Fig 9. SHAPE dataset analysis.** A: Comparison of −1 PRF sequence SHAPE reactivity dataset reported by Manfredonia et al. [13], Huston et al. [12], Yang et al. [46], and Zhang et al. [15]. Reactivity at or below 0.3 is considered to be low or non-reactive indicating the base is paired. B: ShapeKnots' predictions using bootstrapped SHAPE values were obtained 10,000 times and averaged for each index. The mean and variance of the 10,000 predictions for each respective SHAPE dataset are shown here. The y-axis indicates the frequency each nucleotide is predicted as paired. Mean value of 1.0 at a specific position conveys that the nucleotide was predicted as paired for all bootstrapped values, and mean value of 0.0 indicates that the nucleotide was predicted as unpaired for all bootstrapped data. C: Bootstrap procedure was repeated with Shapify used for prediction. Red arrows mark differences from ShapeKnots predictions.

Furthermore, positions 12, 13, 40, 41, 46, 47, and 59 tend to be predicted as paired via ShapeKnots; however, with Shapify these bases are more likely predicted as unpaired. Multiple SHAPE datasets reported reactivity at these positions, indicating Shapify predictions better match the SHAPE data.

ShapeKnots tends to predict loci 61 and 62 as unpaired (consistent with SHAPE datasets), while Shapify predictions tend towards a paired prediction.

In general, Shapify follows the SHAPE datasets more closely and appears more resilient to extreme values in the SHAPE data, seen by the relatively lower variance especially when data by Yang et al. [46] was used.

## Discussion

Here we discuss implications of our SARS-CoV-2 −1 PRF structure prediction results to advance potential treatment development.

### Conserved similarity

First we aligned viral genomes and used RNAz to identify a region of conserved structural similarity among SARS-CoV, SARS-CoV-2, and MERS-CoV. In addition to the expected similarity between SARS-CoV and SARS-CoV-2, we detected structural similarity with MERS-CoV in its −1 PRF pseudoknot region despite low sequence similarity between the two. This result was hypothesized by Fourmy and Yoshizawa [24], and SHAPE data supports the existence of a pseudoknot at the same position in all three coronaviruses [72]. Zhang et al. [15] identified a similar 3D RNA structure using de novo computer modeling in a range of betacoronaviruses including MERS-CoV, SARS-CoV, and SARS-CoV-2. This aligns well with the structural similarity we present in this work. We note that while RNAz detected similarity in the −1 PRF of SARS-CoV-2 and MERS-CoV, its predicted structure is pseudoknot-free whereas the expected structure for this region is indeed a pseudoknotted structure. We therefore, continued our quest to better determine the structure in both cases.

### Structural prediction

By following the hierarchical folding hypothesis we explored the structural landscape of the SARS-CoV-2 and MERS-CoV −1 PRF pseudoknots, expanding on previous structure prediction efforts [15, 24]. Recent work using optical tweezers to identify folding pathways of the SARS-CoV-2 frameshifting pseudoknot strongly supports hierarchical folding, as Stem 1 was identified to fold first followed by Stem 3 [10]. Our predictions for energetically favourable initial stems aligns with this result, with initial stem 1 possessing the lowest free energy by a significant margin (ID 1, cf. Table 4).

In both SARS-CoV-2 and MERS-CoV, the second most energetically favourable initial stem (ID 2, cf. Tables 4 and 6) results in a structure prediction that is markedly different from the native structure (cf. Tables 5 and 7). Initial stem 2's formation is, however, supported by our structural alignment. ShapeKnots predictions identified additional paths from initial stem 2 to native-adjacent structures that intriguingly contained two different sets of pseudoknotted base pairs (kissing-hairpin).

The minimum free energy structure predicted by Iterative HFold for SARS-CoV-2 −1 PRF matched the native structure. The predicted minimum free energy structure for MERS-CoV possesses a recognizable three stemmed structure resembling the native type for the SARS-CoV-2 frameshifting pseudoknot. In addition, the two initial stems (ID 1 and 4, Table 6) that led to this prediction for MERS-CoV, strongly resemble initial stems 1 and 3 (Table 4) that led to native and native-adjacent structures in SARS-CoV-2. The lowest free energy initial stem

(ID 1) for SARS-CoV-2 and MERS-CoV, respectively, is highly conserved in the structural alignment (cf. Table 3). These indicators point to possible kinetic paths for frameshifting pseudoknots that may be consistent across coronaviruses. Indeed, similar structures have been observed to initiate frameshifting in human coronavirus HCV 229E [73]. As a future direction we seek to further expand our structural analysis to additional coronaviruses and upstream targets including attenuator hairpin and palindromic sequences [74].

For SARS-CoV-2 five of the predicted pseudoknotted structures including the native structure, resulted from multiple initial stem input constraints. These structures can be reached from different starting points, which may indicate their increased conformational plasticity. SARS-CoV-2 has greater path resiliency in achieving the native −1 PRF pseudoknot structure and forming pseudoknots in general, when compared with MERS-CoV −1 PRF pseudoknot predictions that were often pseudoknot-free. Pseudoknot predictions for SARS-CoV-2 had lower minimum free energy than those for MERS-CoV. Furthermore, there is a smaller energy gap between the minimum free energy structure of SARS-CoV-2 and the next lowest free energy prediction; where MERS-CoV had a larger gap between the minimum free energy prediction and the next lowest free energy prediction. This may be related to the overall trend that SARS-CoV and SARS-CoV-2 exhibited greater stable RNA structuredness across their genomes compared to MERS-CoV [72]. We hypothesize this may be a factor in the continued spread among various species of COVID-19. Further experiments are needed to confirm the exact structure of non-native conformations for the SARS-CoV-2 −1 PRF pseudoknot and how these specific structures correlate with frameshifting efficiency and viral pathology.

The native pseudoknotted structure for SARS-CoV-2 −1 PRF is formed by crossing Stems 1 and 3 (an H-Type pseudoknot). However, we observed a structure that connects Stems 2 and 3 through a "shifted/smaller" Stem 3 (that crosses Stem 2). This pattern is observed for both SARS-CoV-2 and MERS-CoV. This change of conformation may be functionally important.

## Mutations effect

To explore the effect of point mutations on structure of SARS-CoV-2 −1 PRF, we used an array of most observed mutations in the population as well as a set of previously studied mutations [24]. The predicted secondary structures based on the SARS-CoV-2 −1 PRF mutated sequences are demonstrably different from those for the reference sequence; certain mutations stabilized while others destabilized the initial stems. In particular, U20C, and G29U destabilized initial stem 1, and did not reach the native pseudoknotted structure. Reduction in frameshifting efficiency was experimentally observed by Neupane et al. [25] as a result of U20C mutation. Mutations in Loop 2 (C43U and U47C) were previously found to reduce frameshifting efficiency in the original SARS-CoV [35], although the mechanism is unknown. Interestingly, while there were changes in the predicted secondary structures for the mutated sequences, the function of the SARS-CoV-2 pseudoknot is expected to be conserved for G29U, C43U, U47C, U58C, and C62U [25]. Given that the structure of the pseudoknot with sequence mutations is different but still functional, we hypothesize that the relationship between pseudoknotted structure and function may be more dynamic and flexible than previously expected. In the case of the U20C mutation, we identified a structure that is significantly different from the native expectation and has low free energy (cf. Table H in S1 File, row 3). This native nonadjacent structure is ranked third in terms of free energy, which is not the case for any other mutation. In addition, U20C mutation was the only SARS-CoV-2 sequence that led to a suboptimal structure predicted for initial stem 1. This could be the cause of competition for paths from initial stem 1, which may be functionally crucial. Further experiments are needed to

validate this result and confirm the structural cause for the decrease in −1 PRF efficiency based on the U20C mutation.

Although there are limited MERS-CoV genomic sequences available, one particular mutation (C13479U) was observed in multiple samples. The minimum free energy predicted structure was unchanged due to the mutation, in which the third stem was shifted in multiple predictions to accommodate the mutated base in a larger loop instead. Notably, the conformational switch (shifted Stem 3) as described above is preserved despite the mutation.

## Incorporating SHAPE

To incorporate chemical modification data into our predictions, we introduced Shapify, an algorithm that takes as input an RNA sequence, SHAPE data, and partial structure information and outputs a possibly pseudoknotted RNA secondary structures based on the hierarchical folding hypothesis and guided by SHAPE reactivity score.

Through Shapify predictions we identified paths from initial stems to secondary structures that showcase remarkable redundancy in the mechanism for achieving the native SARS-CoV-2 −1 frameshifting pseudoknot, as well as native-adjacent structures. Shapify's predictions can help capture both the conformational flexibility, diverse structural landscape, and convergence towards native structure of the SARS-CoV-2 −1 PRF pseudoknot.

We observed more consistency in predictions for native and native-adjacent structures: multiple paths from different initial stems to these structures, and agreement when various SHAPE datasets were each used as constraint. Given that the frameshift is vital for viral replication, we expect structural resilience in achieving the frameshift inducing structure. Such structural resiliency can be seen by convergence to the native and native-adjacent structures from different initial stems. Note the four black paths in Fig 6 that indicate the same predicted pseudoknot for each of the four SHAPE datasets as constraint. In further support of the proposed native structure resiliency, is the lack of path redundancy for non-native adjacent predictions (cf. Fig 7). Interestingly, initial stem 2, which is energetically favourable, has multiple paths to different native non-adjacent structures. Further study is required to determine the effect of this mechanism in regulating frameshift efficiency.

Using crystallography for the SARS-CoV-2 −1 PRF pseudoknot accurate to 2.09 angstroms, Roman et al. [48] unveiled marked differences compared to the previously identified crystallography-based structures for the frameshifting structure, including a shorter Stem 2. Here using Shapify we identified multiple new conformations for the SARS-CoV-2 −1 PRF, including a shorter Stem 2, different location for formation of Stem 2 or pseudoknotted base pairs that are not reflected in the native structure.

Recently, using a method combining graph theory, secondary structure prediction, SHAPE structural probing, and thermodynamic ensemble modeling Schlick et al. [14, 16] identified three structural motifs for the SARS-CoV-2 −1 PRF. One of these motifs (denoted as '3_6', coarse grained three stem structure invariant to stem length), corresponds with the first, second, and fourth lowest free energy non-native structures identified by Shapify. Following cryo-EM experiments Bhatt et al. [32] also identified a structure that corresponds with the '3_6' motif.

Our results indicate that NN1, NN2 and NN4 share paths to form following the pairing of initial stems 3, 8, or 14. In addition, both initial stems 8 and 14 had paths to two of the '3_6' motif structures, depending on which SHAPE dataset was used as constraint. In the case of initial stem 8 leading to the formation of the fifth lowest free energy non-native structure, the path is suboptimal, as the initial stem 8 can also lead to the formation of the native structure, the first, or the third lowest free energy non-native structure.

In addition it should be noted that kinetics may also play a role in initial stem formation. Specifically, refolding of RNA after the ribosome passes over and unfolds the pseudoknot may favor pairing towards the 5' end which exits the ribosome first.

Combining our predicted structures in the absence of SHAPE data (using Iterative HFold) for mutated SARS-CoV-2 −1 PRF sequences, and Shapify's predictions with four SHAPE datasets on the reference sequence, we observed that any modification to initial stem 1 (including destabilization or missing initial stem 1 due to mutation) resulted in no native structure prediction, possibly because initial stem 1 is a key path to the native structure (note in [Fig 6] structural path from initial stem 1 to the native structure is supported by all four SHAPE datasets). High degree of agreement among all SHAPE datasets is observed also for initial stems 5, 13, 15 (to NN6) and 16 (to NN19). This may justify the decrease in the −1 PRF efficiency previously observed for U20C in SARS-CoV-2 (possibly due to destabilization of initial stem 1), and C43U (possibly due to destabilization of initial stems 5 and 15) and U47C (possibly due to stabilization of initial stem 13) experimentally observed in the original SARS-CoV virus. This can further point to possible decrease/change in efficiency of −1 PRF in C13U (due to destabilization of initial stem 5 and introduction of a novel stem 13*a*), G29U (due to destabilization of initial stem 1 and introduction of a novel stem 13*b*), and C62U (missing the initial stem 16 and introduction of a novel stem 13*c*). Our predicted structures in the absence of SHAPE reactivity data included no native structure for U20C, G29U and C62U. We hypothesize initial stems 1, 5, 13, 15 and 16 to be important transient structures in the structural path of the −1 PRF of SARS-CoV-2 genome. Further experiments are needed to assess this hypothesis.

## Comparison with ShapeKnots

Comparing performance of Shapify with that of ShapeKnots, a heuristic method that utilizes SHAPE reactivity data, we demonstrated that Shapify identified lower free energy structures than the ones identified by ShapeKnots. ShapeKnots predictions were not found to effectively utilize the initial stems (i.e., ShapeKnots performance was better without the initial stems as constraint). In addition, Shapify predictions better align with the SHAPE datasets for the −1 PRF structure of SARS-CoV-2. Unlike ShapeKnots, Shapify minimizes the free energy of the possibly pseudoknotted output structure relative to the given input structure and the SHAPE reactivity data. Therefore Shapify's method of adding pseudoknotted stems is better motivated energetically than that of ShapeKnots. In addition, Shapify is flexible, allowing minor modification to the input structure to allow formation of energetically favourable base pairs.

## Bootstrap

We observed discrepancies in multiple positions in the secondary structure predictions of ShapeKnots when compared to the SHAPE reactivity data used. For instance, SHAPE datasets indicated high reactivity that was not evident in the actual secondary structure prediction. We therefore performed bootstrapping to assess how much secondary structure prediction for ShapeKnots (and Shapify) are influenced by the accuracy of the given SHAPE reactivity data. For ShapeKnots we observed that approximately half of the nucleotides were not impacted at all by change in the value of SHAPE reactivity induced by the bootstrapping method when Manfredonia et al. and Huston et al. datasets were used. In case of the Yang et al. dataset we observed more variability in the structures predicted after bootstrapping; this can also be attributed to more extreme values in this dataset. Sensitivity to SHAPE data was less severe in structures predicted by Shapify as evaluated by our bootstrapping method. While ShapeKnots

seems to be more sensitive to extreme values in the SHAPE data used, both methods seem to be more influenced by their thermodynamic parameters than by SHAPE data.

We note that *in vivo* SHAPE data is collected by probing the entire 30, 000+ nt SARS-CoV-2 viral genome. If a structure spans across the slippery sequence, it can be difficult to understand how unfolding triggered by ribosome translocation affects formation of the frameshifting pseudoknot and its ability to initiate the frameshift. Here we account for possible inaccuracies in the *in vivo* SHAPE reactivity data by including the *in vitro* SARS-CoV-2 SHAPE dataset from Zhang et al. [15] which focused on a specific fragment of the viral genome, the −1 PRF region, and is corroborated by cryo-EM imaging.

The four SHAPE datasets presented here have some similar trends but are unique at various positions. This could be attributed to the different protocols or approaches for data collection. It may also indicate that the RNA was in different structure conformations for each individual measurement during the high-throughput experiment. Structural probing methods measure multiple molecules and then average the results to obtain final reactivity data, which may lead to noise in the signal when various molecules are each in different structure conformations. Disagreement between SHAPE datasets raises questions to the extent of reliance on using SHAPE data. The SHAPE datasets used in this work, for example, reported significant reactivity for positions 23 and 29 (indicating unpaired region), while both indices are believed to be paired in Stem 1 [10]. Here we acknowledge that tightness of the pseudoknotted loops caused by dimerization with other loops of the structure have been previously cited as possibly affecting correctness of SHAPE data [23]. However, in the SHAPE datasets we analyzed we could not find a non-reactive region (in two or more SHAPE datasets) that covers a loop region of the SARS-CoV-2 −1 PRF native structure.

Previous efforts on RNA 3D modeling of the SARS-CoV-2 −1 PRF pseudoknot acknowledged the relevance of non-native structures and conformational plasticity [6], but this concept has not been fully integrated into the latest prediction efforts [23]. Efforts in identifying unique stable 3D conformations of the SARS-CoV-2 −1 PRF pseudoknot can have a great impact in treatment development [23]. We believe such endeavors can benefit from a more comprehensive overview of the SARS-CoV-2 −1 PRF pseudoknot secondary structure landscape, one that includes non-native structures. Given the complicated nature of modeling tertiary interactions, the initialization of such simulations should account for multiple initial secondary structures of the pseudoknot. Our structural predictions can be utilized in 3D physics-based modeling of pseudoknots as alternative starting points or to provide context to SARS-CoV-2 structure prediction efforts, where small secondary structure differences can have a profound impact on resulting tertiary interactions and three-dimensional conformations of the pseudoknot [17–22].

## Conclusions

We set out to expand knowledge of the SARS-CoV-2 −1 PRF structural conformation to aid with the current efforts for identification of potential treatment options. We detected −1 PRF pseudoknot conserved structural similarity among SARS-CoV, SARS-CoV-2, and MERS-CoV; and discussed varying degrees of RNA structuredness and similar −1 PRF folding paths between the viruses. We identified energetically favourable initial stems for SARS-CoV-2 and MERS-CoV −1 PRF pseudoknots, and used these initial stems as constraint for secondary structure prediction via the Iterative HFold algorithm. To further predict possibly pseudoknotted structures, we introduced Shapify to utilize both SHAPE data and partial structure information. To assess Shapify's predicted structures, we compared them with structures predicted by ShapeKnots; we found Shapify's predicted structures more stable (lower in free

energy) than those predicted by ShapeKnots. In order to understand the effect of SHAPE data on predictions, we performed a bootstrap procedure, which demonstrated Shapify is less sensitive to SHAPE data compared to ShapeKnots. Shapify pseudoknot predictions reveal SARS-CoV-2 possesses consistent path resiliency in achieving the −1 PRF pseudoknot native structure, with multiple pathways from initial stems to native and native-adjacent structures as compared with paths to native non-adjacent structures.

We determined the effects of most observed point mutations to the SARS-CoV-2 and MERS-CoV −1 PRF pseudoknot sequences, finding that certain mutations may stabilize or destabilize pseudoknot structure. Our results indicate SARS-CoV-2 initial stem 1 is critical in formation of the native −1 PRF SARS-CoV-2 pseudoknot, with initial stem 2 having paths to native non-adjacent structures. Understanding structural similarities between coronaviruses such as initial stem/pseudoknot alignment and path convergence can shed light on their mechanisms of function and help with finding effective treatments for existing and emerging contagions. Similarities in structure between coronaviruses may contain vital information that can validate proposed explanations for frameshifting. The individual and consensus structures presented here can further inform structure prediction, providing previously unavailable insights to explain paths of structure formation. We plan to extend this structural analysis to larger frameshifting pseudoknot-adjacent regions in a wider array of coronaviruses.

## Supporting information

**S1 File. Supplementary materials.**
(XLSX)

## Acknowledgments

We thank and acknowledge the Computational Biology Research and Analytics Lab for invaluable feedback.

## Author Contributions

**Conceptualization:** Luke Trinity, Hosna Jabbari, Ulrike Stege.

**Data curation:** Luke Trinity, Lance Lansing.

**Formal analysis:** Luke Trinity.

**Funding acquisition:** Hosna Jabbari, Ulrike Stege.

**Investigation:** Luke Trinity.

**Methodology:** Luke Trinity, Hosna Jabbari, Ulrike Stege.

**Project administration:** Luke Trinity, Hosna Jabbari, Ulrike Stege.

**Software:** Luke Trinity, Ian Wark, Lance Lansing.

**Supervision:** Hosna Jabbari, Ulrike Stege.

**Validation:** Luke Trinity.

**Visualization:** Luke Trinity.

**Writing – original draft:** Luke Trinity, Hosna Jabbari, Ulrike Stege.

**Writing – review & editing:** Luke Trinity, Hosna Jabbari, Ulrike Stege.

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
