## [Decision Letter · Decision Letter 0]

6 Jan 2023

Dear Mr. Trinity,

Thank you very much for submitting your manuscript "Shapify: Paths to SARS-CoV-2 frameshifting pseudoknot" for consideration at PLOS Computational Biology. As with all papers reviewed by the journal, your manuscript was reviewed by members of the editorial board and by several independent reviewers. The reviewers appreciated the attention to an important topic. Based on the reviews, we are likely to accept this manuscript for publication, providing that you modify the manuscript according to the review recommendations.

Thank you for your attention to the comments submitted for the earlier version of your article.

The additional analyses, comparison to other methods, and other revisions have substantially improved the work.

Yet all three reviewers made additional valuable suggestions to improve the work.

Please attend to them in full in your revised paper.

Sincerely,

Tamar Schlick

Academic Editor

PLOS Computational Biology

Nir Ben-Tal

Section Editor

PLOS Computational Biology

Thank you for your attention to the comments submitted for the earlier version of your article.

The additional analyses, comparison to other methods, and other revisions have substantially improved the work.

Yet all three reviewers made additional valuable suggestions to improve the work.

Please attend to them in full in your revised paper.

Reviewer's Responses to Questions

**Comments to the Authors:**

Reviewer #1: The manuscript by Trinity et al., has been updated to reflect comments from other reviewers. It is now a well-expressed description of the Shapify algorithm used to predict RNA structures. The authors use existing RNA chemical reactivity (SHAPE) data and mutational analyses to demonstrate the strengths of their method over the existing Shapeknots method for analyzing the frameshift-stimulating pseudoknots of betacoronaviruses. RNA structures are not static, especially those processed by the ribosome, and the authors do a great job of presenting the possible structures elucidated in the pipeline they describe in the context of the known biology.

There are two minor edits I would recommend prior to publication:

First, the phrase 'increased virility' in line 604 is not quite appropriate. I would recommend something like 'continued spread among various species'.

Second, the authors mention kissing-hairpin structures in a couple of places (lines 506 and 578) that could be further investigated in the future. While these structures haven't been shown to facilitate frameshifting for the betacoronaviruses (SARS, MERS) analyzed in this work, similar structures that facilitate frameshifting been described for the alphacoronavirus 229E frameshift (Herold & Siddell, 1993, NAR 21:5838-42). It would not be out of place if the authors choose to add the analysis of elaborate internal structures present in larger frameshift-stimulating pseudoknots of the alphacoronaviruses to their future directions (line 588).

Reviewer #2: In this work, Trinity et al. applied Iterative HFold to study SARS-CoV-2 -1 programmed ribosomal frameshifting (-1 PRF) structure, and adapted the program to develop Shapify by incorporating SHAPE reactivity data to guide hierarchical folding.

To expand the knowledge of SARS-CoV-2 frameshifting pseudoknot, the authors evaluated structural similarities between -1 PRF structures of SARS-CoV-2 and MERS-CoV, with a focus on predicted initial stems and suboptimal secondary structures. The authors also examined the effects of mutations on the -1 PRF structures from selected SARS-CoV-2 and MERS-CoV-2 mutated sequences, which exist in the population and have been validated experimentally for their effect on -1 PRF.

The authors applied the Shapify pipeline to study the SARS-CoV-2 -1 PRF structure, obtained an ensemble of suboptimal structures and identified folding paths. For Shapify development, the parameters of the pseudo energy term from SHAPE reactivity data was determined from 30 RNAs with known structures and SHAPE datasets. The authors compared the performance of Shapify and the existing method ShapeKnots using four available SARS-CoV-2 SHAPE datasets (in vivo and in vitro).

The authors revised the manuscript based on previous reviewers’ feedback, including comparisons with existing method ShapeKnots and published results of -1 PRF structures. The Shapify pipeline is novel and unique in providing possible paths to the final folding. The scope of the study to identify notable non-native secondary structures and elucidate the structure-function relationship regarding -1 PRF is promising. However, these strengths could be better explained with improved figures and analyses. The manuscript seems to reach a publishable state.

Below are the detailed comments:

1.The structural paths are presented in Figures 6 and 7 from initial stems to final foldings. The native pseudoknot 2D structure in Figure 6 seems to be a combination of initial stems 1, 2, and 3. It seems each of the three stems still exist in the final 2D structure that it reaches. Interestingly, initial stems 1 and 3 lead to the native or native-adjacent structure, but initial stem 2 reaches native non-adjacent structures. The finding might correspond to the -1 PRF mechanism, as it suggests possible folding pathways of the H-type pseudoknot. The authors mentioned related folding pathway study and initial stem 2’s formation in Discussion. It might be helpful to elaborate the findings in this work and relate the paths in Figures 6 and 7 to biological functions.

2. Some initial stems in Table 4 might be located similarly and share common residues, for example, Stem 8 and 9. However, they might lead to different secondary structures in Figures 6 and 7. Does this suggest any base pair that might be critical to the stem in the final folding? It might be helpful to compare how base pairs in the initial stems are preserved in the final 2D structure.

3. Using initial stems as constraints in ShapeKnots for paths might not be helpful, as the method is not the same with Shapify. The authors did mention this in their response to previous Reviewer 1 comment 3.

Reviewer #3: I think the authors have addressed the scientific concerns that were raised in the previous rounds of review adequately. The clarity of the writing has been improved, but overall the manuscript still remains somewhat difficult to read. One thing that may help to ensure that the message of the paper is as clear as possible is to separate "discussion" from "conclusions", instead of combining them as done here. Please first discuss the implications of the results (along with any caveats and subtleties), and only after the discussion is finished present a brief (1-2 paragraphs) summary of the key results along with any concluding remarks in a separate "Conclusions" section.

Some minor points:

Line 5: Although ref. 3 reviews therapeutic approaches for Covid, it does not, in fact, discuss PRF as a therapeutic target, which is how it is used as a citation. The authors should cite a review that explicitly discusses targeting PRF therapeutically, some examples are Kelly et al. Virology (2021) (in the context specifically of coronaviruses) and Anokhina & Miller Acc Chem Res (2021) (in a broader context).

Line 303: under "Bootstrapping", I would recommend the authors avoid saying that they used a bootstrap "to generate additional information based on each original dataset", because bootstrapping cannot actually generate new information from a dataset!

respectively

**Have the authors made all data and (if applicable) computational code underlying the findings in their manuscript fully available?**

Reviewer #1: Yes

Reviewer #2: Yes

Reviewer #3: None

PLOS authors have the option to publish the peer review history of their article (what does this mean?). If published, this will include your full peer review and any attached files.

Reviewer #1: No

Reviewer #2: No

Reviewer #3: No

Figure Files:

Data Requirements:

Reproducibility:

References:

---

## [Editor Report · Decision Letter 1]

5 Feb 2023

Dear Mr. Trinity,

We are pleased to inform you that your manuscript 'Shapify: Paths to SARS-CoV-2 frameshifting pseudoknot' has been provisionally accepted for publication in PLOS Computational Biology.

Best regards,

Tamar Schlick

Academic Editor

PLOS Computational Biology

Nir Ben-Tal

Section Editor

PLOS Computational Biology

---

## [Editor Report · Acceptance letter]

23 Feb 2023

PCOMPBIOL-D-22-01468R1 

Shapify: Paths to SARS-CoV-2 frameshifting pseudoknot

Dear Dr Trinity,

I am pleased to inform you that your manuscript has been formally accepted for publication in PLOS Computational Biology. Your manuscript is now with our production department and you will be notified of the publication date in due course.

With kind regards,

Zsofi Zombor
